# Sleep EEG in young people with 22q11.2 deletion syndrome: A cross-sectional study of slow-waves, spindles and correlations with memory and neurodevelopmental symptoms

Nicholas A Donnelly[1,2]*[†], Ullrich Bartsch[3,4][†], Hayley A Moulding[5], Christopher Eaton[5], Hugh Marston[4], Jessica H Hall[5], Jeremy Hall[5], Michael J Owen[5], Marianne BM van den Bree[5][‡], Matt W Jones[6][‡]

[1]Centre for Academic Mental Health, University of Bristol, Bristol, United Kingdom; [2]Avon and Wiltshire Partnership NHS Mental Health Trust, Avon, United Kingdom; [3]School of Physiology, Pharmacology and Neuroscience, University of Bristol, Bristol, United Kingdom; [4]Translational Neuroscience, Eli Lilly, Windlesham, United States; [5]Medical Research Council Centre for Neuropsychiatric Genetics and Genomics, Cardiff University, Cardiff, United Kingdom; [6]University of Bristol, Bristol, United Kingdom

*For correspondence:
nick.donnelly@bristol.ac.uk

[†]These authors contributed equally to this work
[‡]These authors also contributed equally to this work

## Abstract

**Background:** Young people living with 22q11.2 Deletion Syndrome (22q11.2DS) are at increased risk of schizophrenia, intellectual disability, attention-deficit hyperactivity disorder (ADHD) and autism spectrum disorder (ASD). In common with these conditions, 22q11.2DS is also associated with sleep problems. We investigated whether abnormal sleep or sleep-dependent network activity in 22q11.2DS reflects convergent, early signatures of neural circuit disruption also evident in associated neurodevelopmental conditions.

**Methods:** In a cross-sectional design, we recorded high-density sleep EEG in young people (6–20 years) with 22q11.2DS (n=28) and their unaffected siblings (n=17), quantifying associations between sleep architecture, EEG oscillations (spindles and slow waves) and psychiatric symptoms. We also measured performance on a memory task before and after sleep.

**Results:** 22q11.2DS was associated with significant alterations in sleep architecture, including a greater proportion of N3 sleep and lower proportions of N1 and REM sleep than in siblings. During sleep, deletion carriers showed broadband increases in EEG power with increased slow-wave and spindle amplitudes, increased spindle frequency and density, and stronger coupling between spindles and slow-waves. Spindle and slow-wave amplitudes correlated positively with overnight memory in controls, but negatively in 22q11.2DS. Mediation analyses indicated that genotype effects on anxiety, ADHD and ASD were partially mediated by sleep EEG measures.

**Conclusions:** This study provides a detailed description of sleep neurophysiology in 22q11.2DS, highlighting alterations in EEG signatures of sleep which have been previously linked to neurodevelopment, some of which were associated with psychiatric symptoms. Sleep EEG features may therefore reflect delayed or compromised neurodevelopmental processes in 22q11.2DS, which could inform our understanding of the neurobiology of this condition and be biomarkers for neuropsychiatric disorders.

**Funding:** This research was funded by a Lilly Innovation Fellowship Award (UB), the National Institute of Mental Health (NIMH 5UO1MH101724; MvdB), a Wellcome Trust Institutional Strategic Support Fund (ISSF) award (MvdB), the Waterloo Foundation (918-1234; MvdB), the Baily Thomas Charitable Fund (2315/1; MvdB), MRC grant Intellectual Disability and Mental Health: Assessing Genomic Impact on Neurodevelopment (IMAGINE) (MR/L011166/1; JH, MvdB and MO), MRC grant Intellectual Disability and Mental Health: Assessing Genomic Impact on Neurodevelopment 2 (IMAGINE-2) (MR/T033045/1; MvdB, JH and MO); Wellcome Trust Strategic Award 'Defining Endophenotypes From Integrated Neurosciences' Wellcome Trust (100202/Z/12/Z MO, JH). NAD was supported by a National Institute for Health Research Academic Clinical Fellowship in Mental Health and MWJ by a Wellcome Trust Senior Research Fellowship in Basic Biomedical Science (202810/Z/16/Z). CE and HAM were supported by Medical Research Council Doctoral Training Grants (C.B.E. 1644194, H.A.M MR/K501347/1). HMM and UB were employed by Eli Lilly & Co during the study; HMM is currently an employee of Boehringer Ingelheim Pharma GmbH & Co KG. The views and opinions expressed are those of the author(s), and not necessarily those of the NHS, the NIHR or the Department of Health funders.

## Editor's evaluation

The authors quantified sleep oscillations and their coordination in young people with 22q11.2 Deletion Syndrome and their siblings. This was done to identify potential biomarkers of later neurodevelopmental diagnoses in 22q11.2 Deletion Syndrome. The core findings based on solid data demonstrate that sleep rhythms in 22q11.2DS are altered in comparison to the control group, as is their relationship with the behavioral expressions of memory consolidation. These are important findings as they directly provide a link between genes and sleep rhythms and memory consolidation.

## Introduction

22q11.2 microdeletion syndrome (22q11.2DS) is caused by a deletion spanning a ~2.6 megabase region on the long arm of chromosome 22. It occurs in ~1:3000–4000 births and is associated with increased risk of neuropsychiatric conditions including intellectual disability, autism spectrum disorder (ASD), attention-deficit hyperactivity disorder (ADHD), and epileptic seizures. (*Cunningham et al., 2018*; *Eaton et al., 2019*; *Moulding et al., 2020*; *Niarchou et al., 2014*). 22q11.2DS is also considered to be one of the largest biological risk factors for schizophrenia, with up to 41% of adults with 22q11.2DS having psychotic disorders (*Karayiorgou et al., 1995*; *Monks et al., 2014*; *Schneider et al., 2014*). However, the neurobiological mechanisms underlying psychiatric symptoms in 22q11.2DS remain unclear. Deep phenotyping of young people with 22q11.2DS may allow their elucidation and therefore enable early detection and/or intervention.

The electroencephalogram (EEG) recorded during non-rapid eye movement (NREM) sleep features spindle and slow-wave (SW) oscillations: highly conserved and non-invasively measurable signatures of neuronal network activity generated by corticothalamic circuits (*Adamantidis et al., 2019*). The properties and co-ordination of these oscillations are candidate biomarkers of brain dysfunction in neuropsychiatric disorders (*Ferrarelli and Tononi, 2017*; *Gardner et al., 2014*; *Manoach et al., 2016*).

Sleep EEG features are altered across many neurodevelopmental disorders, including schizophrenia, including first episode psychosis, as well as first degree relatives (*Chouinard et al., 2004*; *Cohrs, 2008*; *Ferrarelli et al., 2007*; *Ferrarelli et al., 2010*; *Göder et al., 2014*; *Bartsch et al., 2019*; *Demanuele et al., 2017*; *Wamsley et al., 2012*; *Manoach and Stickgold, 2019*; *Castelnovo et al., 2018*; *Keshavan et al., 1998*); ADHD (*Cortese et al., 2009*; *Gorgoni et al., 2020*; *Lunsford-Avery et al., 2016*), ASD although findings have been inconsistent (*Gorgoni et al., 2020*; *Lehoux et al., 2019*) and a range of rare genetic conditions, including Down syndrome, Fragile-X syndrome and Angelman syndrome (*Angriman et al., 2015*).

We have recently shown that the majority of young people with 22q11.2DS have sleep problems, particularly insomnia and sleep fragmentation, that associate with psychopathology (*Moulding et al., 2020*). However, this analysis was based on parental report; the neurophysiological properties of sleep in this condition remain unexplored. Furthermore, it has been demonstrated that neuroanatomical features associated with psychopathology in 22q11.2DS significantly converge with those in

idiopathic psychiatric disorders (*Ching et al., 2020*). Therefore, studying the sleep EEG in 22q11.2DS may produce insights that can be generalized to broader populations, affording a unique opportunity to clarify the relationship between sleep EEG and psychiatric risk.

We hypothesized that 22q11.2DS would be associated with alterations in sleep EEG features relative to controls, including altered spindle and SW events, and aberrant spindle-SW coupling. We investigated these hypotheses in a cross-sectional study of young people with 22q11.2DS and unaffected sibling controls, combining detailed neuropsychiatric assessments with overnight high-density EEG recordings and a sleep-dependent memory task.

## Results

### Psychopathology and sleep architecture in 22q11.2 DS

Young people living with 22q11.2DS (n=28) and healthy control siblings (n=17) completed semi-structured research diagnostic interviews to quantify Full Spectrum Intelligence Quotient (FSIQ), neuropsychiatric symptoms and self- and carer-reported sleep behavioral problems (*Table 1* and *Figure 1—figure supplement 1*). Participants with 22q11.2DS had a lower mean FSIQ (reported as Odds Ratio (OR) or group difference (GD) with [95% confidence interval]): FSIQ, GD = − 28.70 [- 40.48, – 16.92], p<(0.001), and higher incidence of anxiety (OR = 3.10 [1.93, 4.99], p<0.001), ADHD (OR = 9.46 [5.12 – 17.48], p<0.001) and ASD symptoms (Odds Ratio [OR]=7.46 [4.76, 11.70], p<0.001), but did not show significantly more psychotic experiences than controls (OR = 4.05 [0.67, 43.67], p = 0.096). Details of the specific psychotic symptoms reported are shown in *Table 2*.

Participants with 22q11.2DS also experienced more sleep problems (OR = 6.27 [2.12, 18.56], p=0.001); more sleep problems were associated with younger age, 22q11.2DS genotype and anxiety symptoms but not with gender, family income, psychotic experiences, ADHD, or ASD symptoms (*Table 3*).

Participants were asked to perform a delayed recall 2D object location task (*Figure 1A*) to test sleep-dependent memory consolidation. Of 42 participants who engaged in the task, those with 22q11.2DS needed more training cycles to reach a 30% performance criterion (Hazard Ratio [95% CI]=0.328 [0.151, 0.714], p=0.005, *Figure 1B*, *Table 4*) and made fewer correct responses in the morning test session (OR = 0.631 [0.45, 0.885], p=0.008, *Figure 1C*, *Table 4*). However, there was no difference between groups in overnight change in correct responses between the evening learning session and the morning test session (*Figure 1D*, *Table 4*). Additionally, there was no association between task performance or accuracy in the morning test session and any psychiatric measure, or FSIQ (*Table 4*).

All participants completed one night of full polysomnography with 64-channel high density EEG recorded at their home. After expert sleep scoring, we compared sleep architecture between 22q11.2DS and controls (*Figure 1E* and *Table 1*). There was no difference in gross measures of sleep such as Total Sleep Time and Sleep Efficiency, suggesting that our EEG recordings did not disrupt sleep differently between groups. However, 22q11.2DS was associated with a reduced percentage of N1 (GD = −2.71 [-5.05,–0.36], p = 0.044) and REM sleep (GD = −4.20 [-7.10,–1.30], p = 0.012) while the percentage of N3 sleep was increased (GD = 5.47 [1.98, 8.96], p = 0.009). There were no significant relationships between sleep architecture metrics and psychiatric measures or FSIQ in 22q11.2DS (*Table 5*).

### Altered spectral properties of the sleep EEG in 22q11.2DS

Given the above evidence for an altered overall distribution of sleep stages in 22q11.2DS, we next used spectral analyses to quantify sleep EEG oscillations in our sample.

Before analyzing all 60 EEG electrodes, we calculated power spectral density (PSD) across frequencies from 0.5 to 20 Hz for controls and in 22q11.2DS for electrode Cz, as both spindle and slow wave oscillations can be reliable detected at this location (*Figure 2A*). We found that power in lower frequencies appeared to be increased in 22q11.2DS across N2 and N3 as well as across a range of frequencies during REM sleep (cluster-corrected p<0.05).

To investigate potential changes in specific oscillatory components of the EEG (particularly at the sigma and SO frequency bands), we first z-scored raw EEG recordings in the time domain to eliminate broadband power differences between recordings, and again compared the PSD (*Figure 2B*). This

**Table 1.** Psychiatric characteristics and sleep architecture.

| Variable | Group | | Type | Statistic (95% CI) | p-value |
|---|---|---|---|---|---|
| | **22q11.2DS, n=28** [a] | **Sibling Control, n=17** [a] | | | |
| Age @ EEG | 14.6 (3.4) | 13.7 (3.4) | Group Difference (22q - Sib) [b] | 0.897 [-1.219, 3.013] | 0.397 |
| Sex | | | Chi-Squared [c] | 0 | 1 |
| Female | 14 (50%) | 9 (53%) | | | |
| Male | 14 (50%) | 8 (47%) | | | |
| Sleep Problem | 1.32 (1.70) | 0.24 (0.56) | Odds Ratio [d] | 6.269 [2.118, 18.556] | **0.001** |
| FSIQ | 76 (13) | 105 (27) | Group Difference (22q - Sib) [e] | −28.696 [-40.478,−16.915] | **<0.001** |
| *missing* | 0 | 1 | | | |
| Anxiety Symptoms | 5.0 (7.8) | 1.4 (2.8) | Odds Ratio [d] | 3.101 [1.929, 4.986] | **<0.001** |
| ADHD Symptoms | 6.0 (6.0) | 0.7 (2.1) | Odds Ratio [d] | 9.456 [5.117, 17.475] | **<0.001** |
| ASD Symptoms | 11 (6) | 1 (2) | Odds Ratio [d] | 7.463 [4.762, 11.697] | **<0.001** |
| *missing* | 1 | 1 | | | |
| Psychotic Experiences | | | Odds Ratio [f] | 4.047 [0.698, 43.668] | 0.096 |
| No PE | 18 (64%) | 15 (88%) | | | |
| PE | 10 (36%) | 2 (12%) | | | |
| N1 (%) | 10.4 (4.7) | 13.6 (4.3) | Group Difference (22q - Sib) [e] | −2.707 [-5.05,−0.363] | **0.044** |
| N2 (%) | 26.2 (8.2) | 27.1 (5.9) | Group Difference (22q - Sib) [e] | −1.089 [-5.146, 2.967] | 0.620 |
| N3 (%) | 30 (7) | 25 (6) | Group Difference (22q - Sib) [e] | 5.473 [1.984, 8.962] | **0.009** |
| REM (%) | 14.4 (4.6) | 18.2 (5.6) | Group Difference (22q - Sib) [e] | −4.198 [-7.1,−1.296] | **0.012** |
| N1 Latency (Minutes) | 23 (18) | 21 (9) | Group Difference (22q - Sib) [e] | 3.486 [-5.538, 12.509] | 0.470 |
| REM Latency (Minutes) | 143 (69) | 140 (49) | Group Difference (22q - Sib) [e] | 9.368 [-19.312, 38.048] | 0.549 |
| Sleep Efficiency (%) | 88 (8) | 89 (9) | Group Difference (22q - Sib) [e] | −1.845 [-5.826, 2.136] | 0.398 |
| Total Sleep Time (Minutes) | 456 (122) | 485 (79) | Group Difference (22q - Sib) [e] | −27.206 [-88.489, 34.077] | 0.413 |
| Awakenings (n) | 42 (52) | 42 (40) | Group Difference (22q - Sib) [e] | 3.097 [-19.732, 25.925] | 0.802 |

[a] Mean (SD); n (%)

[b] Linear Model

[c] Pearson's Chi Squared Test

[d] Generalised Linear Mixed Model

[e] Linear Mixed Mode

[f] Fisher's Exact Test

**Table 2.** Psychotic experiences details.

Frequency of specific psychotic experiences

| Type of PE | 22q11.2DS | Sibling |
|---|---|---|
| Unusual thought content/ Delusional ideas | 8 | 1 |
| Suspiciousness/ Persecutory ideas | 5 | 0 |
| Grandiose Ideas | 3 | 2 |
| Perceptual Abnormalities/ Hallucinations | 8 | 2 |
| Disorganised communication | 4 | 0 |

Count of total distinct types of psychotic experience

| Number of PE | 22q11.2DS | Sibling |
|---|---|---|
| 0 | 18 | 15 |
| 1 | 2 | 0 |
| 2 | 2 | 1 |
| 3 | 2 | 1 |
| 4 | 4 | 0 |

Details of psychotic experiences reported by participants with 22q11.2DS and unaffected sibling controls in the CAPA interview.

analysis revealed reduced relative power in the sigma frequency band in 22q11.2DS in N2 and N3 sleep (cluster-corrected $P<0.05$).

Next, we used irregular-resampling auto-spectral analysis (*Hahn et al., 2020*; *Wen and Liu, 2016*) to separate the oscillatory and fractal (1 /f) components of the EEG. This analysis demonstrated that the power of the fractal component of the PSD was increased in 22q11.2DS across a wide range of frequencies in N2, N3, and REM sleep (*Figure 2C*). However, in the oscillatory component of the EEG we found that power in the sigma band appeared to be reduced in 22q11.2DS (*Figure 2D*) but to have a higher peak frequency. Every participant had a distinct peak in oscillatory activity in the sigma frequency band (*Figure 2—figure supplement 1A*).

We then focused on a set of PSD derived measures from N2 and N3 sleep: power in the slow delta (<1.25 Hz) and sigma (10–16 Hz) bands, and peak sigma frequency. Additionally, we calculated the y-intercept (cons) and the negative exponent (beta) of a 1 /f line fit to the fractal component of the signal to allow comparison of non-oscillatory activity between groups, for N2, N3, and REM epochs. We extracted these measures across all 60 EEG electrodes and fitted generalized additive mixed models to the data from all electrodes for each measure. *Table 6* shows all spectral EEG measures calculated. A detailed topographical analysis revealed that 22q11.2DS showed lower sigma power, but higher sigma frequency during N2 and N3 sleep in central regions, and higher total power, as indexed by the 1 /f intercept measure across N2, N3, and REM sleep, particularly in fronto-lateral regions (*Figure 2E–G*). In contrast, there were no substantial differences in slow delta power or 1 /f slope between groups.

Individual data and group boxplots for this set of spectral measures extracted at electrode Cz are shown in *Figure 2—figure supplement 1B*, and plots of spectral measures with age are shown in *Figure 2—figure supplement 1C*, demonstrating clear positive relationships between age and sigma frequency, and negative relationships between age and the overall PSD power (constant) and slope (beta), as previously demonstrated (*Hahn et al., 2020*). Group average topoplots for all PSD derived measures are shown in *Figure 2—figure supplement 2*

## Spindles and slow waves in 22q11.2DS

To further interrogate the thalamocortical oscillations underlying genotype-dependent alterations in spectral power and frequency, we quantified individual spindle and slow wave (SW) events using automated detection algorithms. For spindle detection, for each participant and each electrode, we individualized the frequency used for spindle detection, using the peak sigma band frequency from our spectral analysis.

*Figure 3A and B* show example spectrograms from electrode Cz for a pair of siblings; one control (*Figure 3A*), one with 22q11.2DS (*Figure 3B*), with detected spindle and SW events overlaid. As expected, these plots clearly indicate the presence of co-occurring spindle and SW events during NREM sleep.

**Table 3.** CAPA sleep problem adjusted model.

| Term | | Odds ratio | p-value |
|---|---|---|---|
| | Sibling | *Reference* | |
| Genotype | 22q11.2DS | 7.867 [1.71, 36.186] | 0.008 |
| | Female | *Reference* | |
| Gender | Male | 1.557 [0.486, 4.986] | 0.456 |
| Age @ EEG | | 0.757 [0.622, 0.921] | 0.005 |
| | <19,999 | *Reference* | |
| | 20,000–39,999 | 0.38 [0.068, 2.13] | 0.271 |
| | 40,000–59,999 | 0.227 [0.034, 1.505] | 0.124 |
| Family income (£PA) | >60,000 | 0.297 [0.043, 2.058] | 0.219 |
| Anxiety symptoms[a] | | 1.117 [1.031, 1.21] | 0.007 |
| ADHD symptoms[a] | | 1.025 [0.945, 1.112] | 0.546 |
| ASD symptoms[a] | | 0.964 [0.869, 1.07] | 0.488 |
| | No PEs | *Reference* | |
| Psychotic experiences (PEs) | PEs | 1.369 [0.646, 2.9] | 0.413 |

[a]Continuous variables (no reference category)

*Associations between CAPA sleep problem count and group, demographic, family and psychiatric covariates, modeled with a generalized linear mixed model, with a poisson distribution and family identity as a random (varying) intercept. Data shown are odds ratios and the 95% confidence interval.*

The average waveforms of spindle and SW events detected on electrode Cz are shown in *Figure 3C* and *Figure 3D*, exemplifying group differences in spindle and SW properties: participants with 22q11.2DS showed increased spindle amplitude across fronto-lateral regions, with accompanying increases in spindle density and frequency across smaller regions (*Figure 3E*). SW amplitude was also increased in central, frontal and lateral areas, but there were no differences in SW density or duration between groups (*Figure 3F*). Individual data from all participants for the measured spindle and SW properties are presented in *Figure 3—figure supplement 1*, and group topoplots for each property in *Figure 3—figure supplement 2*. *Figure 3—figure supplement 3* shows SW-triggered potentials across the scalp.

## Increased spindle-SW coupling in 22q11.2DS

The relative timing of spindle and SW events is coupled during NREM sleep, and thought to reflect limbic-thalamic-cortical interactions (*Bartsch et al., 2019*; *Demanuele et al., 2017*; *Djonlagic et al., 2021*; *Helfrich et al., 2018*; *Latchoumane et al., 2017*). An illustrative example of an overlapping spindle and SW detection is shown in *Figure 4A*. To investigate whether 22q11.2DS was associated with alterations in spindle-SW coupling, we first calculated the proportion of spindles that overlapped a detected SW (where a spindle peak fell within +/-1.5 seconds of a detected SW negative peak). We then calculated the SW phase angle at the point of peak amplitude in each spindle, and the mean resultant length [MRL, a measure of the circular concentration of phase angles, with greater values indicating a more consistent spindle-SW phase relationship (*Djonlagic et al., 2021*)] at the peak of each spindle.

The mean angle of spindle-SW coupling showed a non-uniform distribution, confirming that spindles tended to consistently occur at particular SW phases (*Figure 4B* shows coupling angles for spindles detected at electrode Cz); at electrode Cz both control and 22q11.2DS participants had significantly non-uniform distributions of spindle-SW coupling phase angles (siblings: mean angle = 27.6°, SD = 0.959, Rayleigh Test for non—uniformity statistic = 0.632, p<0.001; 22q11.2DS: mean

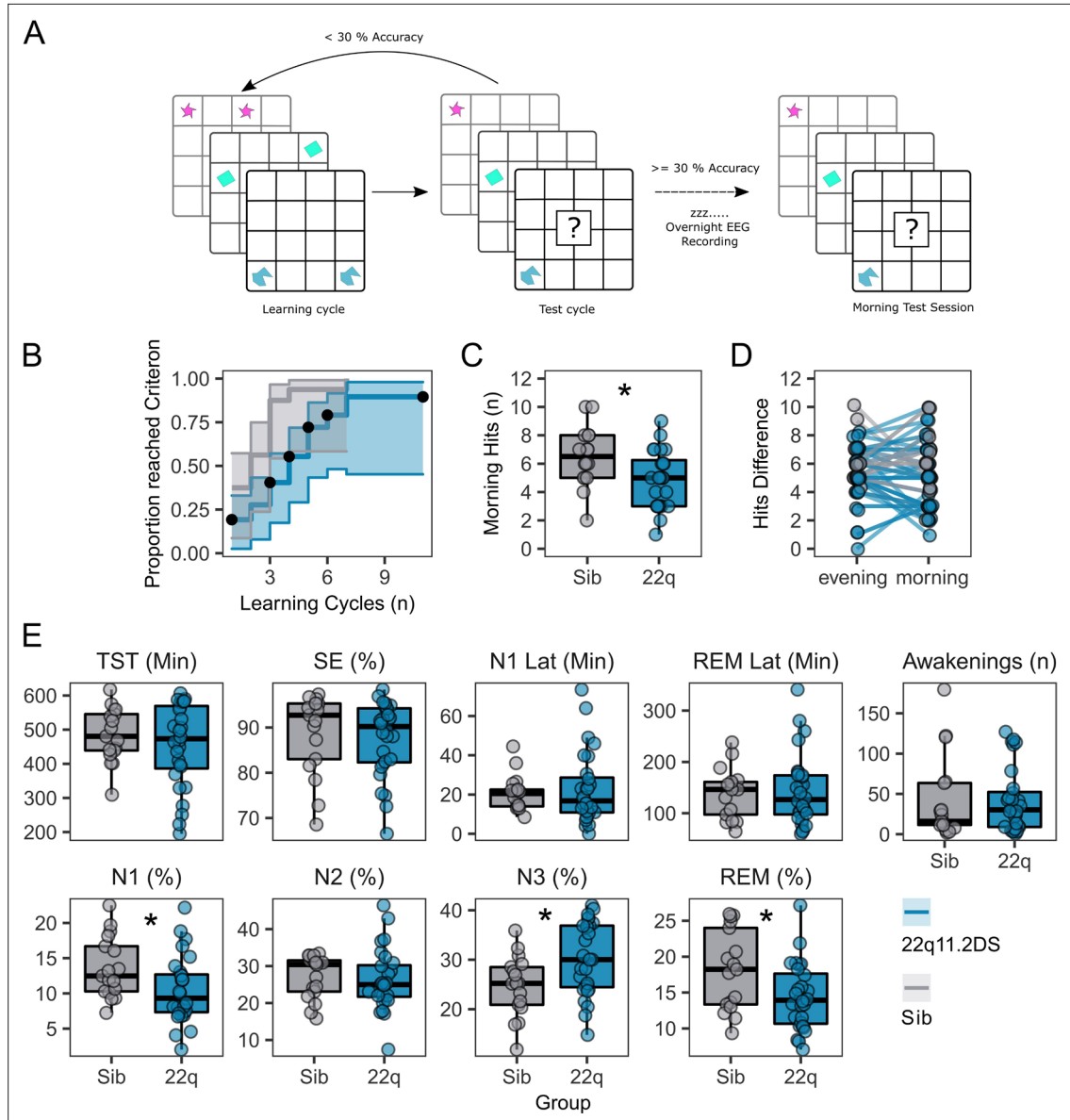

**Figure 1.** Memory task performance and sleep architecture features of 22q11.2DS.

(**A**): Schematic of the 2D object location task. The evening before sleep EEG recordings, participants first were sequentially presented with pairs of images on a 5 x 6 grid. In a subsequent test cycle, they were presented with one image of the pair, and were required to select the grid location of the other half of the pair. If the participant did not achieve > 30% accuracy, they would have another learning cycle. In the morning a single test cycle was undertaken. (**B**): Plot of performance in acquiring the 2D object location task, showing the proportion of participants in each group reaching the 30% performance criterion after each learning cycle. Shaded areas represent the 95% confidence interval. Black dots show when participants were right-censored due to stopping the task prior to reaching the 30% criterion. (**C**): Box plots of performance in the morning test session, where participants had one cycle of the memory task. Number of correct responses is out of a possible 15. Asterix indicate the group difference is statistically significant, generalised linear mixed model, p<0.05 (see *Table 2* for full statistics). (**D**): Plots of change in performance between the final evening learning session and the morning test session. Each participant is represented as a point, with a line connecting their evening and morning performance. Points have been slightly jittered to illustrate where multiple participants had the same score. (**E**): Box and whisker plots showing sleep architecture features: Total sleep time (TST) in minutes, Sleep efficiency (SE) as a percentage, Latency to N1 sleep (minutes), Latency to first REM sleep (minutes), Number of awakenings after sleep onset (n), Percentage of hypnogram in N1 sleep, Percentage of hypnogram in N2 sleep, Percentage of hypnogram in N3 sleep, and Percentage of hypnogram in REM sleep. Asterixes indicate the group difference is statistically significant, linear mixed model, *P*<0.05 (see *Table 1* for full statistics). Boxes represent the median and IQR, with the whiskers representing 1.5 x the IQR. Individual participant data are shown as individual points. Points have been slightly jittered in the x direction only to illustrate where multiple participants had similar results.

The online version of this article includes the following figure supplement(s) for figure 1:

**Figure supplement 1.** Individual Psych Hypno Data.

**Table 4.** Memory task acquisition and test session performance.

Cycles to Criterion Cox Model

| Term | | Hazard ratio | p-value |
|---|---|---|---|
| Group | Control | *Reference* | |
| | 22q11.2DS | 0.328 [0.151, 0.714] | **0.005** |
| Gender | Female | *Reference* | |
| | Male | 1.389 [0.642, 3.005] | 0.400 |
| Age @ EEG | | 1.029 [0.91, 1.164] | 0.650 |

Cycles to Criterion Cox Model – Adjusted for Psychiatric Measures - 22q11.2DS Only

| Term | | Hazard Ratio | p-value |
|---|---|---|---|
| Gender | Female | *Reference* | |
| | Male | 2.314 [0.542, 9.882] | 0.257 |
| Psychotic experiences | No PEs | *Reference* | |
| | PEs | 0.203 [0.041, 1.012] | 0.052 |
| Age @ EEG | | 1.139 [0.933, 1.390] | 0.200 |
| FSIQ | | 1.026 [0.972, 1.082] | 0.355 |
| Anxiety symptoms | | 0.992 [0.879, 1.120] | 0.900 |
| ADHD symptoms | | 0.915 [0.760, 1.102] | 0.349 |
| ASD symptoms | | 1.027 [0.926, 1.139] | 0.616 |

Morning Accuracy Binomial Model

| Term | | OR | p-value |
|---|---|---|---|
| Group | Control | *Reference* | |
| | 22q11.2DS | 0.631 [0.45, 0.885] | **0.008** |
| Gender | Female | *Reference* | |
| | Male | 1.083 [0.762, 1.538] | 0.657 |
| Age @ EEG | | 0.997 [0.945, 1.051] | 0.900 |

Morning Accuracy Binomial Model - Adjusted for Psychiatric Measures - 22q11.2DS Only

| Term | | OR | p-value |
|---|---|---|---|
| Gender | Female | *Reference* | |
| | Male | 1.623 [0.807, 3.268] | 0.174 |
| Psychotic experiences | No PEs | *Reference* | |
| | PEs | 0.556 [0.296, 1.032] | 0.065 |
| Age @ EEG | | 1.012 [0.924, 1.108] | 0.803 |
| FSIQ | | 1.004 [0.982, 1.027] | 0.716 |

*Table 4 continued on next page*

*Table 4 continued*

Cycles to Criterion Cox Model

| | | |
|---|---|---|
| Anxiety symptoms | 1.028 [0.969, 1.091] | 0.353 |
| ADHD symptoms | 0.973 [0.924, 1.023] | 0.288 |
| ASD symptoms | 1.018 [0.973, 1.066] | 0.441 |

Evening – Morning Difference

| Term | | Group Difference | p-value |
|---|---|---|---|
| Group | Control | *Reference* | |
| | 22q11.2DS | −0.424 [-1.923, 1.074] | 0.565 |
| Gender | Female | *Reference* | |
| | Male | −0.5 [-2.036, 1.035] | 0.512 |
| Age @ EEG | | −0.023 [-0.256, 0.21] | 0.839 |

Associations between genotype group, sex, age and psychiatric symptoms and performance in the 2D object location task.

angle = 9.51°, SD = 0.769, Rayleigh test statistic 0.744, p<0.001), but no difference in coupling angle was observed between groups: Watson Williams test $F_{1,43}$ = 1.341, p = 0.253.

We then compared coupling between groups across all electrodes. Compared to siblings, 22q11.2DS was not associated with any change in the proportion of spindles overlapping SW, but was associated with increased MRL across a central region, indicating less variable spindle-SW phase coupling (*Figure 4C*). There were only minor differences in preferred coupling angle in 22q11.2DS (*Figure 4D*). Per participant data for the overlap and MRL measures recorded on electrode Cz are presented in *Figure 4—figure supplement 1* and group topoplots in *Figure 4—figure supplement 2*. SW-Triggered scalograms, showing the location of spindle-frequency acitivty relative to the SW waveform, are presented in *Figure 4—figure supplement 3*.

## Sleep feature associations with memory recall

Next, we tested whether features of the sleep EEG which demonstrated significant group differences (REM 1 /f intercept, spindle amplitude, SW amplitude and spindle-SW MRL), interacted with group effects on accuracy in the morning test session. As there were no group differences in change in task performance overnight, but groups differed in morning test performance, we focused on number of correct responses in the morning memory test session. For features extracted at electrode Cz (*Figure 5A*), significant features x genotype interactions were observed (all *P*<0.05). Applying the same analysis across all recording electrodes (*Figure 5B*), we found significant clusters of negative interactions between group and REM intercept, spindle and SW amplitude across multiple central and posterior electrodes: higher spindle and SW amplitudes were associated with higher accuracy in controls; in 22q11.2DS, higher amplitudes were associated with lower accuracy. We did not observe any interaction between spindle-SW MRL and task performance.

## Mediation of genotype effects on psychiatric symptoms by EEG features

Finally, we used mediation models to investigate whether the effects of 22q11.2DS genotype on psychiatric symptoms and FSIQ were potentially mediated via sleep EEG measures (*Figure 6A*); such mediation would support the potential role for quantitative sleep EEG measures serving as biomarkers for psychiatric disorders e.g. (*Manoach and Stickgold, 2019*). We calculated the total effect of genotype on psychiatric measures and IQ, the indirect (mediated) effect of EEG measures, and the proportion of the total effect that may be mediated by EEG measures, correcting for multiple comparisons

**Table 5.** Regression of sleep architecture features in 22q11.2DS.

| Measure | Variable | Beta (95% CI) | Adjusted P-value (BH) |
|---|---|---|---|
| | Sex | −0.059 [-5.21, 5.092] | 0.981 |
| | Age @ EEG | 0.101 [-0.808, 1.01] | 0.963 |
| | CAPA sleep problems | 0.003 [-1.632, 1.639] | 0.963 |
| | FSIQ | 0.135 [-0.044, 0.313] | 0.963 |
| | Anxiety symptoms | 0.158 [-0.38, 0.696] | 0.963 |
| | ADHD symptoms | −0.288 [-0.682, 0.105] | 0.963 |
| | ASD symptoms | 0.33 [-0.006, 0.666] | 0.963 |
| N1 (%) | Psychotic experiences | −2.758 [-6.921, 1.404] | 0.963 |
| | Sex | 2.183 [-8.465, 12.831] | 0.963 |
| | Age @ EEG | 0.097 [-1.782, 1.976] | 0.915 |
| | CAPA sleep problems | −0.407 [-3.787, 2.974] | 0.915 |
| | FSIQ | 0.195 [-0.174, 0.564] | 0.915 |
| | Anxiety symptoms | 0.254 [-0.859, 1.366] | 0.915 |
| | ADHD symptoms | −0.691 [-1.505, 0.123] | 0.915 |
| | ASD symptoms | 0.28 [-0.414, 0.974] | 0.915 |
| N2 (%) | Psychotic experiences | −3.603 [-12.208, 5.002] | 0.915 |
| | Sex | −0.849 [-10.545, 8.847] | 0.915 |
| | Age @ EEG | 0.675 [-1.037, 2.386] | 0.915 |
| | CAPA sleep problems | 1.399 [-1.68, 4.477] | 0.997 |
| | FSIQ | −0.062 [-0.398, 0.273] | 0.997 |
| | Anxiety symptoms | −0.43 [-1.442, 0.583] | 0.816 |
| | ADHD symptoms | 0.359 [-0.382, 1.1] | 0.816 |
| | ASD symptoms | −0.05 [-0.682, 0.582] | 0.997 |
| N3 (%) | Psychotic experiences | 3.852 [-3.984, 11.688] | 0.816 |
| REM (%) | Sex | 2.516 [-2.744, 7.775] | 0.816 |
| | Age @ EEG | −0.682 [-1.61, 0.246] | 0.816 |
| | CAPA sleep problems | −1.168 [-2.837, 0.502] | 0.816 |
| | FSIQ | 0.138 [-0.044, 0.32] | 0.235 |
| | Anxiety symptoms | 0.732 [0.182, 1.281] | 0.421 |
| | ADHD symptoms | −0.295 [-0.697, 0.107] | 0.788 |
| | ASD symptoms | −0.054 [-0.397, 0.288] | 0.235 |
| | Psychotic experiences | −0.404 [-4.655, 3.847] | 0.719 |

*Table 5 continued on next page*

*Table 5 continued*

| Measure | Variable | Beta (95% CI) | Adjusted P-value (BH) |
|---|---|---|---|
| | Sex | –8.061 [-32.213, 16.092] | 0.235 |
| | Age @ EEG | –1.225 [-5.487, 3.037] | 0.235 |
| | CAPA sleep problems | 0.484 [-7.184, 8.152] | 0.947 |
| | FSIQ | –0.237 [-1.073, 0.6] | 0.235 |
| | Anxiety symptoms | –0.62 [-3.143, 1.902] | 0.638 |
| | ADHD symptoms | 0.143 [-1.703, 1.989] | 0.638 |
| | ASD symptoms | –0.194 [-1.769, 1.38] | 0.638 |
| N1 Latency (Minutes) | Psychotic experiences | 1.894 [-17.625, 21.414] | 0.107 |
| | Sex | –30.174 [-110.781, 50.433] | 0.638 |
| | Age @ EEG | 1.517 [-12.707, 15.741] | 0.638 |
| | CAPA sleep problems | 10.761 [-14.83, 36.353] | 0.638 |
| | FSIQ | –2.491 [-5.283, 0.301] | 0.638 |
| | Anxiety symptoms | –2.763 [-11.183, 5.656] | 0.638 |
| | ADHD symptoms | 3.909 [-2.251, 10.069] | 0.254 |
| | ASD symptoms | –2.116 [-7.371, 3.139] | 0.254 |
| REM Latency (Minutes) | Psychotic experiences | –14.904 [-80.048, 50.24] | 0.363 |
| | Sex | 1.813 [-8.177, 11.802] | 0.254 |
| | Age @ EEG | –0.099 [-1.862, 1.663] | 0.872 |
| | CAPA sleep problems | –0.979 [-4.151, 2.192] | 0.256 |
| | FSIQ | 0.286 [-0.06, 0.632] | 0.254 |
| | Anxiety symptoms | 0.587 [-0.456, 1.63] | 0.254 |
| | ADHD symptoms | –0.671 [-1.435, 0.092] | 0.256 |
| | ASD symptoms | 0.444 [-0.207, 1.096] | 0.483 |
| Sleep Efficiency (%) | Psychotic experiences | –1.347 [-9.42, 6.726] | 0.736 |
| Total Sleep Time (Minutes) | Sex | 76.874 [-63.554, 217.302] | 0.87 |
| | Age @ EEG | –14.689 [-39.469, 10.092] | 0.87 |
| | CAPA sleep problems | –13.13 [-57.714, 31.453] | 0.87 |
| | FSIQ | 0.156 [-4.708, 5.021] | 0.736 |
| | Anxiety symptoms | 9.132 [-5.536, 23.799] | 0.507 |
| | ADHD symptoms | –10.338 [-21.069, 0.394] | 0.87 |
| | ASD symptoms | –0.955 [-10.11, 8.2] | 0.507 |
| | Psychotic experiences | –121.448 [-234.938,–7.958] | 0.772 |

*Table 5 continued on next page*

*Table 5 continued*

| Measure | Variable | Beta (95% CI) | Adjusted P-value (BH) |
|---|---|---|---|
| | Sex | 5.695 [-44.381, 55.77] | 0.772 |
| | Age @ EEG | 0.932 [-7.904, 9.769] | 0.772 |
| | CAPA sleep problems | 4.938 [-10.96, 20.836] | 0.844 |
| | FSIQ | −1.403 [-3.138, 0.332] | 0.844 |
| | Anxiety symptoms | −2.383 [-7.613, 2.848] | 0.844 |
| | ADHD symptoms | 2.443 [-1.383, 6.27] | 0.844 |
| | ASD symptoms | −2.331 [-5.596, 0.933] | 0.336 |
| Awakenings (n) | Psychotic experiences | −15.59 [-56.06, 24.879] | 0.772 |

Associations between sleep architecture measures (proportion of N1, N2, N3 and REM sleep, latency to N1 and REM sleep, Sleep Efficiency, Total Sleep Time and total Awakenings), sex, age and psychiatric and cognitive (FSIQ) covariates, in participants with 22q11.2DS. Regression models were fit with linear mixed models, with family identity as a random (varying) intercept. Data presented are regression beta coefficients with 95% confidence intervals.

using cluster-based permutation testing. The largest effects were of mediation of genotype effects on anxiety and ADHD symptoms by SW amplitude and spindle – SW coupling, with mediation of genotype effects on ADHD symptoms by REM constant, and genotype effect on ASD symptoms by spindle amplitude (*Figure 6B* and *Table 7*). There was little evidence for consistent mediation of genotype effects on sleep problems, psychotic experiences or FSIQ.

## Discussion

### Summary of findings

We performed an analysis of sleep EEG characteristics in 22q11.2DS, correlating these with psychiatric symptoms, sleep architecture, and performance in a memory task.

Our previous results, based on primary carer reports, discovered sleep disruption, particularly insomnia and restless sleep in 22q11.2DS (*Moulding et al., 2020*), which was associated with psychopathology. We extend these findings to show that, compared to unaffected control siblings, 22q11.2DS is associated with decreased N1 and REM sleep, increased N3 sleep, increased overall EEG power and altered power and frequency in the sigma band during NREM sleep.

These finding were accompanied by changes in NREM sleep-related events: increased spindle amplitude, density, and frequency; increased SW amplitude; and increased spindle-SW phase coupling. The relationship between spindle and SW features and performance of a spatial memory task differed by group, with a positive correlation between spindle and SW amplitude and performance in controls, but a negative relationship in 22q11.2DS. Finally, group differences in anxiety, ADHD, and ASD symptoms were mediated by several EEG measures, particularly SW amplitude and spindle – SW coupling, across multiple electrodes.

### Relationship to previous findings – spindles and spindle-slow wave coupling

We observed increased spindle amplitude, frequency, and density in 22q11.2DS, accompanied by increased spindle-SW coupling, and evidence that spindle amplitude and spindle-SW coupling mediated genotype effects on anxiety, ADHD, and ASD symptoms.

The clinical presentation of 22q11.2DS is heterogenous (*Cunningham et al., 2018*), including anxiety, ADHD and ASD symptoms, reduced IQ and increased risk of psychotic disorders (*Schneider et al., 2014*). Adult schizophrenia is consistently associated with reduced spindle activity (*Ferrarelli et al., 2007*; *Ferrarelli et al., 2010*; *Lai et al., 2022*), a finding replicated in a study of early onset schizophrenia (*Gerstenberg et al., 2017*), and meta-analytic evidence suggests spindle deficits increase with higher symptom burden and longer illness duration (*Lai et al., 2022*). However, increased

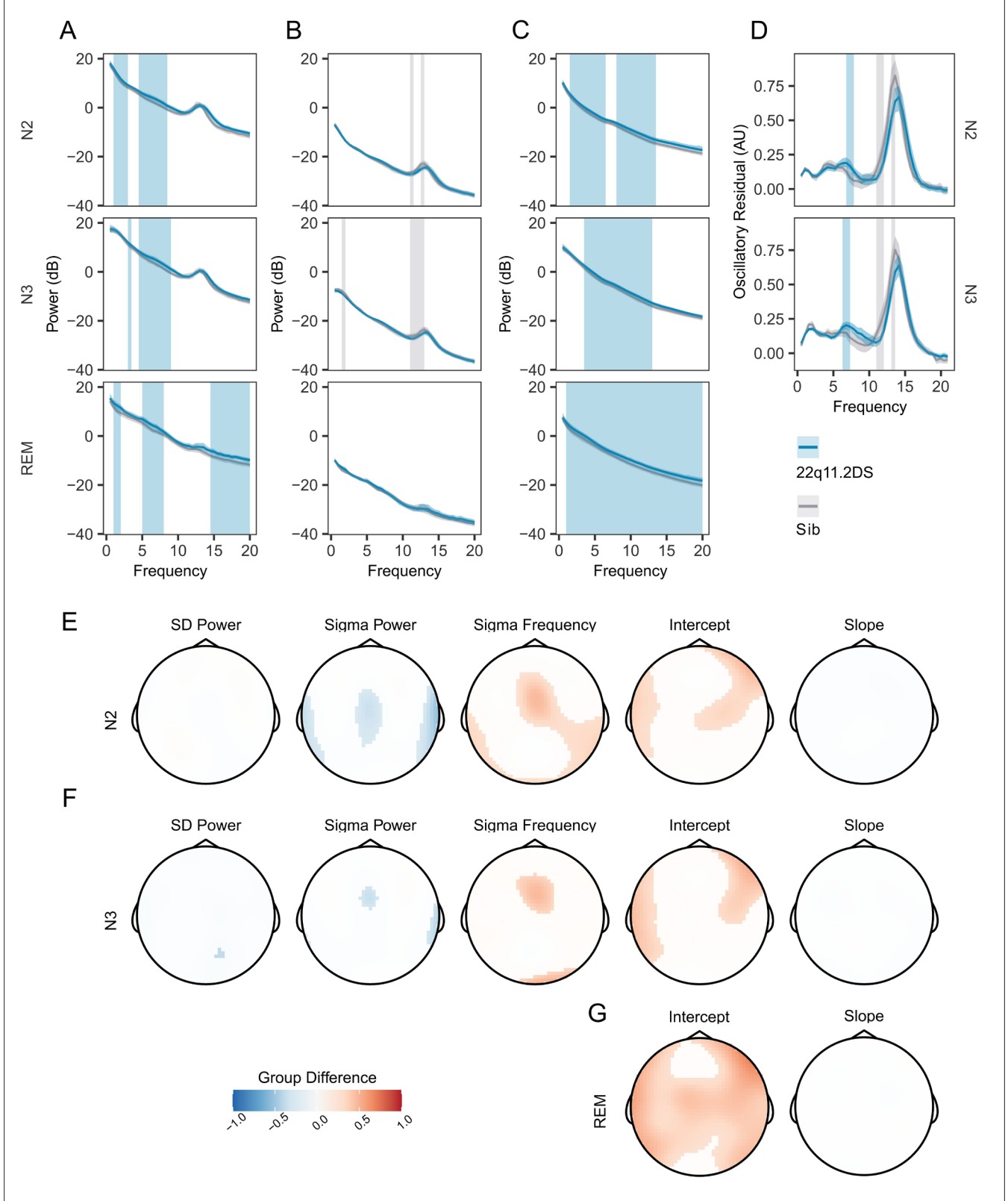

**Figure 2.** Increased PSD power and Sigma Frequency in 22q11.2DS. (**A**) *Raw Welch Power Spectral Density (PSD, in decibels, 10 * log₁₀ of the PSD) on electrode Cz across Stage N2, N2, and REM sleep. Lines show group mean power (blue = 22q11.2DS, gray = Sibling), with bootstrapped 95% confidence intervals of the mean. Patches show regions of significant (cluster corrected) difference between groups (blue = 22q11.2DS >Sibling; grey = 22q11.2DS <Sibling), with 22q11.2DS being associated with increased power at lower frequencies. (**B**) Welch PSD of Z-Scored EEG signals*

*Figure 2 continued on next page*

*Figure 2 continued*

*on electrode Cz, as in (**A**); with 22q11.2DS being associated with lower power in the sigma frequency band (10–16 Hz) (**C**) Fractal (1 /f) component of EEG signal processed using the IRASA method on electrode Cz, conventions as (**A**). Higher power across a wide frequency range in 22q11.2DS. (**D**) Oscillatory component of the EEG signal processed using the IRASA method on electrode Cz, conventions as (**A**). (**E**) Topoplots of group difference calculated from multilevel generalized additive models fit to the full 60 channel dataset for the five measures (mean Slow Delta power, mean Sigma power and peak Sigma frequency, 1 /f Intercept and 1 /f Slope) recorded in N2 sleep. Positive differences represent z score group differences indicate 22q11.2DS >Sibling (red colors); negative group differences (blue colors) indicate 22q11.2DS <Sibling. Only regions were where the probability of direction statistic for group difference was >0.995 are colored. (**F**) As in (**E**), for N3 sleep. (**G**) As in (**F**), for REM sleep. Note as REM sleep lacks prominent oscillatory activity, we have not calculated models for SD or sigma related measures in REM as these would not be meaningful.*

The online version of this article includes the following figure supplement(s) for figure 2:

**Figure supplement 1.** Individual PSDs.

**Figure supplement 2.** Group PSD Topos.

spindle amplitudes and densities have been observed in healthy adolescents with raised polygenic risk scores for schizophrenia (*Merikanto et al., 2019*). In contrast, no clear differences in spindle properties have been found in other 22q11.2DS-associated neurodevelopmental disorders such as ADHD (*Prehn-Kristensen et al., 2011*) and ASD (*Maski et al., 2007*).

**Table 6.** EEG measure summary.

| Measure group | Measure details | Sleep stage |
|---|---|---|
| Spectral | Mean Slow Delta Power | N2 |
| Spectral | Mean Slow Delta Power | N3 |
| Spectral | Mean Sigma Power | N2 |
| Spectral | Mean Sigma Power | N3 |
| Spectral | Peak Sigma Frequency | N2 |
| Spectral | Peak Sigma Frequency | N3 |
| Spectral | Aperiodic Signal Slope | N2 |
| Spectral | Aperiodic Signal Slope | N3 |
| Spectral | Aperiodic Signal Slope | REM |
| Spectral | Aperiodic Signal Intercept | N2 |
| Spectral | Aperiodic Signal Intercept | N3 |
| Spectral | Aperiodic Signal Intercept | REM |
| Spindle | Density | N2 +N3 |
| Spindle | Amplitude | N2 +N3 |
| Spindle | Frequency | N2 +N3 |
| Slow Wave | Density | N2 +N3 |
| Slow Wave | Amplitude | N2 +N3 |
| Slow Wave | Duration | N2 +N3 |
| Spindle – Slow Wave Coupling | Spindle – Slow Wave Overlap (z-scored against shuffled data) | N2 +N3 |
| Spindle – Slow Wave Coupling | Spindle – Slow Wave Mean Resultant Length (z-scored against shuffled data) | N2 +N3 |
| Spindle – Slow Wave Coupling | Spindle – Slow Wave Mean Coupling Angle | N2 +N3 |

All derived EEG measures, grouped by signal type spectral, derived from the PSD; spindle, derived from individual detected spindle events; slow wave, derived from individual detected slow wave events and measures derived from spindle – slow wave coupling.

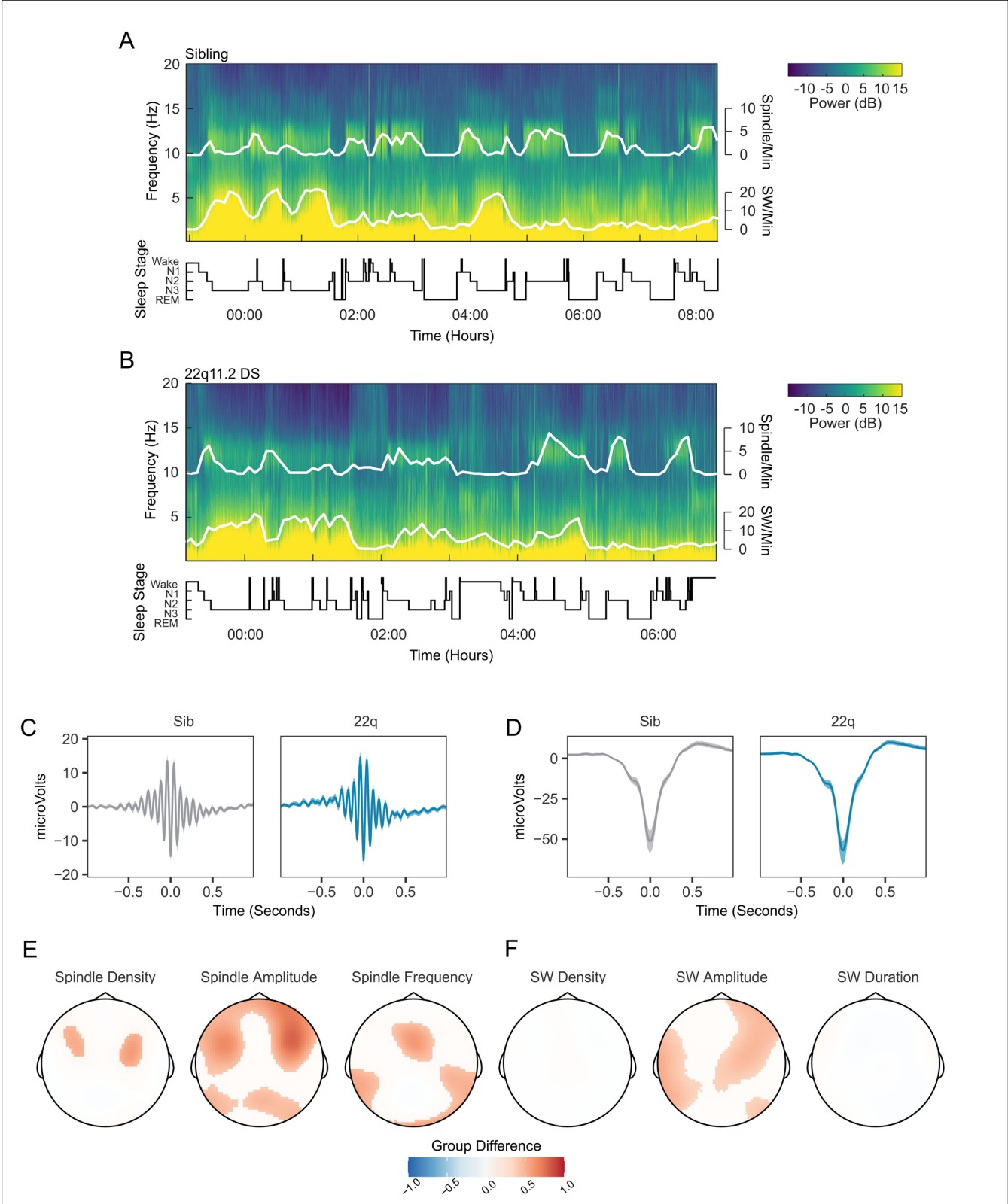

**Figure 3.** Spindles and slow waves in 22q11.2DS. (**A**) *Example spectrogram of a whole night EEG recording from electrode Cz for an example sibling. The associated hypnogram is displayed below the spectrogram in black, detected spindle and slow wave events are overplotted in white. The co-occurrence of spindle events with epochs of N2 sleep, and of SW events and N3 sleep can be observed.* (**B**) *Example spectrogram of a whole night EEG recording from electrode Cz for an example participant with 22q11.2DS, sibling of the participant illustrated in A* (**C**) *Average spindle waveforms*

*Figure 3 continued on next page*

Figure 3 continued

detected on electrode Cz for siblings (left, gray), and 22q11.2DS (right, blue). For each individual the average spindle waveform at Cz was calculated, these averaged waveforms were then calculated for all siblings or all participants with 22q11.2DS. Shaded areas highlight the bootstrapped 95% confidence interval of the mean. (**D**) Average SW waveforms detected on electrode Cz, same conventions as C (**E**) Topoplots of group differences in spindle density, amplitude and frequency, Z-transformed, across all 60 electrodes, from GAMM analyses. Only regions with significant group differences are highlighted. Red colors indicate values of the parameter of interest are greated in 22q11.2DS; blue color that the parameter of interest is greater in siblings (**F**) Topoplots of group differences in SW density, amplitude and duration, conventions as in E.

The online version of this article includes the following figure supplement(s) for figure 3:

**Figure supplement 1.** Individual event data.

**Figure supplement 2.** Group event topoplots.

**Figure supplement 3.** SW-triggered potentials.

Spindle properties and spindle – SW relationships change across the lifespan (*Djonlagic et al., 2021*; *Hahn et al., 2020*; *Hahn et al., 2022*; *Purcell et al., 2017*; *Zhang et al., 2021*): spindle density, power and frequency increases from childhood to adolescence alongside spindle-SW coupling, while power in lower frequency declines (*Hahn et al., 2020*; *Jenni and Carskadon, 2004*; *Tarokh and Carskadon, 2010*). Our findings could therefore be interpreted in the context of alterations in developmental processes in 22q11.2DS: higher spindle amplitude, density, and frequency in young, at-risk populations for psychosis could mark an aberrant maturational state, which leads to reduced spindle activity in adulthood for individuals who go on to develop psychotic disorders, with higher symptom burden and illness duration linked to greater reductions in spindle activity.

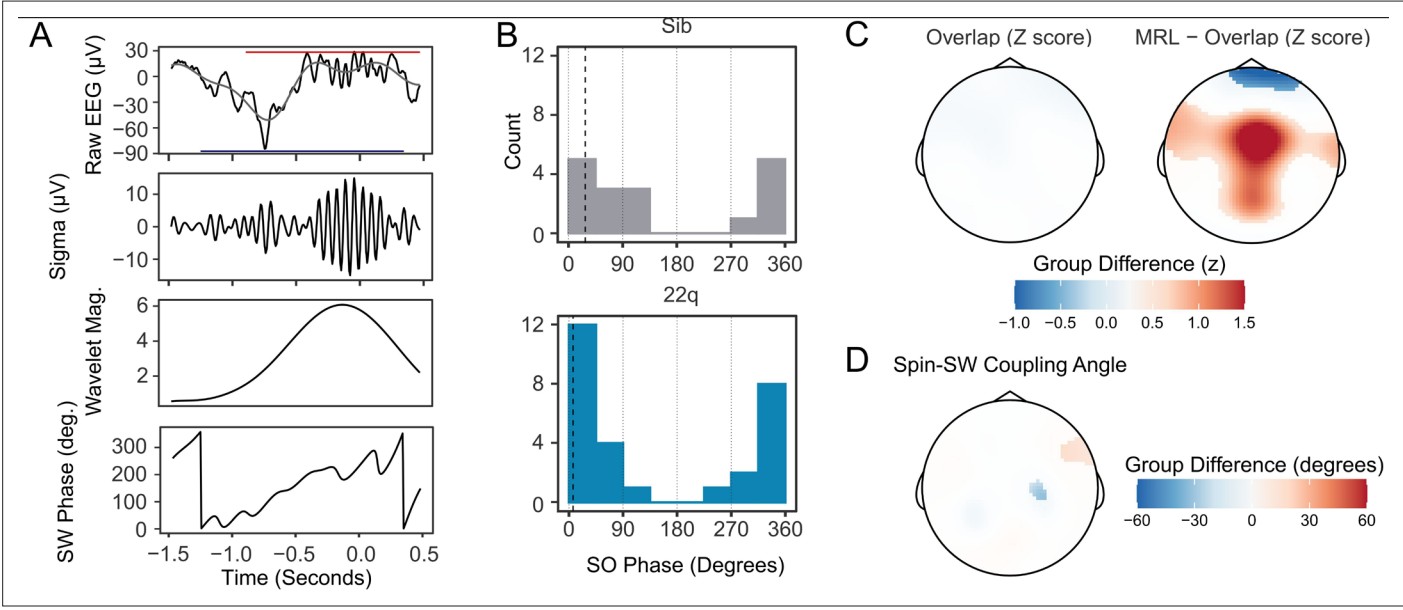

**Figure 4.** Increased spindle-SW coupling in 22q11.2DS. (**A**) Illustrative plot of a single spindle and SW recorded at electrode Cz in a control sibling. From top to bottom, panels show the raw EEG (black) with Slow-Wave frequency (0.25–4 Hz) filtered data superimposed (gray) and with the detected boundaries of the spindle and SW highlighted with a red and blue horizontal bar, the sigma-filtered raw signal (10–16 Hz); the magnitude of the continuous wavelet transform of the signal (center frequency 13 Hz); and the SW phase (in degrees). (**B**) Histograms of the mean SW phase angle of spindles detected overlapping an SW for all participants at electrode Cz. The SO phase angles are as defined in (**A**). Black vertical dashed lines indicate the mean coupling phase angle for each group (**C**) Topoplots of group difference in spindle-SW coupling properties: z-transformed spindle-SW overlap (left), and z-transformed mean resultant length (right). The color represents the difference in z-score between groups where a multilevel generalized additive model fit to each dataset predicts a difference between group. (**D**) Topoplots of mean Spindle-SW coupling phase angle, where a multilevel generalized additive model fit to each dataset predicts a difference in coupling phase angle between groups.

The online version of this article includes the following figure supplement(s) for figure 4:

**Figure supplement 1.** Individual coupling data.

**Figure supplement 2.** Topoplots of group average values for spindle – SW coupling measures (overlap, MRL and mean angle).

**Figure supplement 3.** SW-Triggered Scalograms.

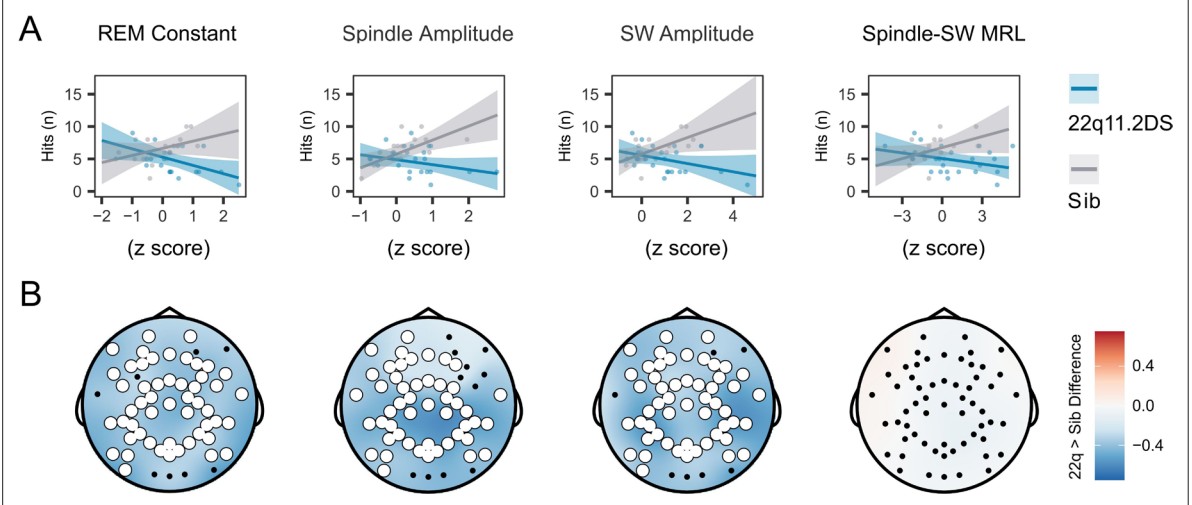

**Figure 5.** EEG signatures of sleep dependent memory consolidation. (**A**) Scatter plot of the relationship between EEG measures (recorded on electrode Cz) and hits in the memory task test session, by group. Lines represent predicted mean values, with 95% confidence interval, from linear mixed model. (**B**) Topoplots of the value of the group*EEG feature interaction term, for models fit to hits in the morning test session. Electrodes highlighted in white indicate a significant interaction for an EEG measure detected on that channel, after correction for multiple comparisons. Note all topoplots are on the same color scale.

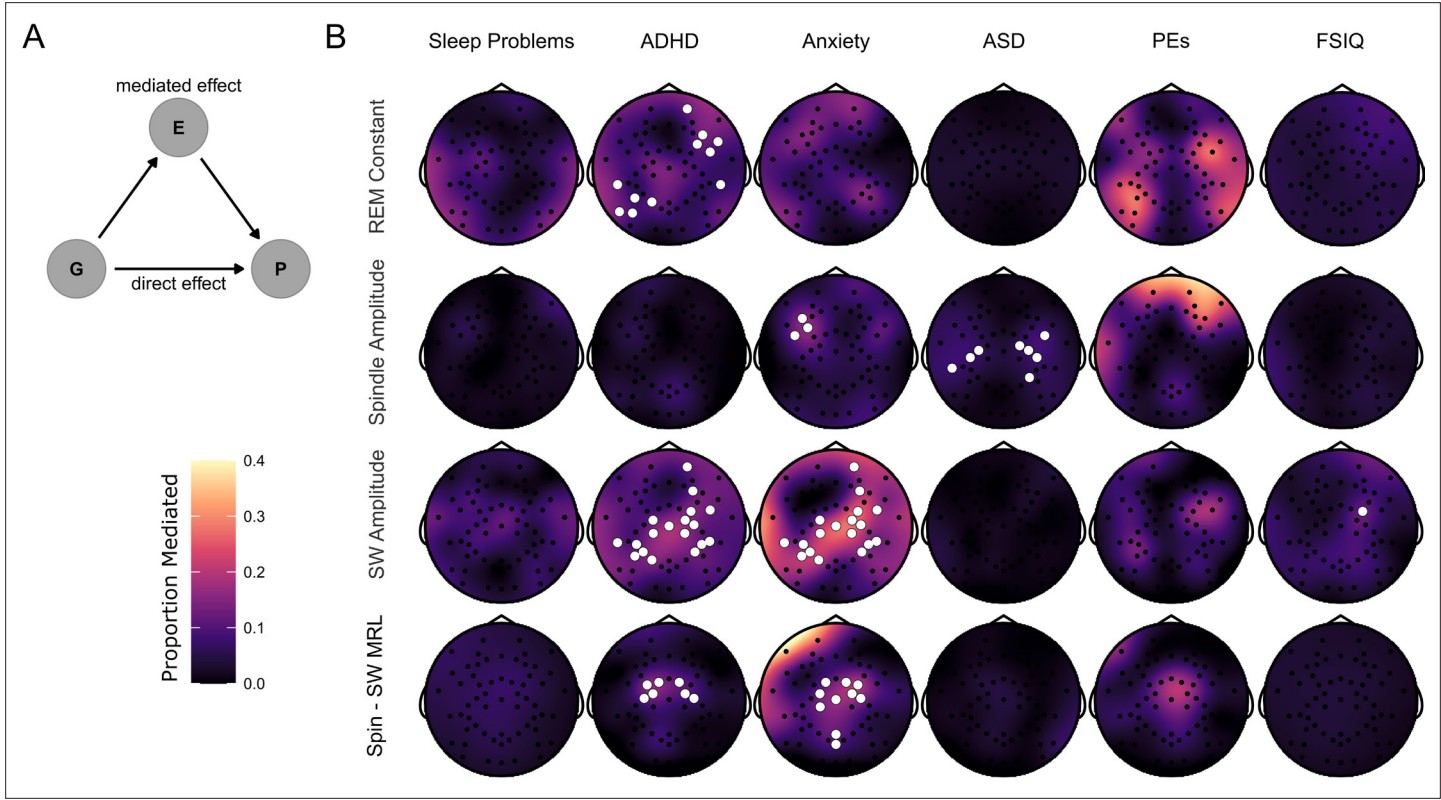

**Figure 6.** Mediation of psychiatric symptoms and FSIQ by sleep EEG features. (**B**) Topoplots of the proportion of the effect of genotype on psychiatric measures and FSIQ mediated by one of four NREM sleep EEG features (REM constant spindle amplitude, SW amplitude and spindle-SW MRL). Fill color represents the Proportion Mediated. Electrodes are highlighted in white where a mediation model fit on data from that electrode had a significant mediated effect and a significant total effect, corrected for multiple comparisons by the cluster method. (**A**) Directed acyclic graph describing the mediation model fit to EEG data. The effect of Group (**G**) on psychiatric measures and FSIQ (**P**) was hypothesized to be mediated by (**E**) – sleep EEG features.

**Table 7.** Average proportions of genotype effects on psychiatric measures and IQ mediated by sleep EEG measures.

| Measure | Mediator | Proportion mediated |
|---|---|---|
| ADHD symptoms | REM constant | 0.11 (0.02) |
| ADHD symptoms | SW amplitude | 0.14 (0.03) |
| ADHD symptoms | Spin - SW MRL | 0.16 (0.03) |
| Anxiety symptoms | Spindle amplitude | 0.17 (0) |
| Anxiety symptoms | SW amplitude | 0.21 (0.05) |
| Anxiety symptoms | Spin - SW MRL | 0.19 (0.07) |
| ASD symptoms | Spindle amplitude | 0.08 (0.02) |
| FSIQ | SW amplitude | 0.18 |

*Proportions of genotype effect on psychiatric measures and FSIQ mediated (Measure) by select sleep EEG features (Mediator) of for all electrodes in significant clusters. Data shown are mean (SD). Note FSIQ does not have an SD as there was only one electrode in a significant cluster.*

In our hands, spindle events peak between ~ 270/-90 and 90 degrees in the SW cycle, with the average coupling angling being early on the first descending part of the negative half-wave, around 10 - 30 degrees, near the trough of the SW (in our frame of reference, 0 degrees is assigned to the positive to negative zero crossing). This peak coupling angle was somewhat different to previous studies e.g. *Hahn et al., 2020*; *Helfrich et al., 2018*, which have found the peak angle of spindle-SW coupling to be between 90-270 degrees in our reference, near the positive peak of the SW.

It has been suggested that "slow" spindles (frequency ~9 – 12 Hz) peak prior to the SW trough (90 degrees in our reference), in contrast to "fast" spindles (frequency > 12 Hz), which peak around the SW peak (~270 degrees, *Mölle et al., 2011*; *McConnell et al., 2021*). We detected spindles using a wavelet-based method where the wavelet centre-frequency was individualised based on each participants sigma frequency PSD, finding each participant had a unimodal distribution of sigma power, rather than a separate 'fast' and 'slow' peak, with the peak frequency being substantially affected by participant age. It is therefore possible that our detected spindle events predominantly reflect events that others have labelled "slow spindles", therefore explaining our observed preferred spindle-SW coupling angle falling on the SW descending phase.

An interesting line of enquiry for future studies with larger datasets would be to explore whether 22q11.2DS, or other neurodevelopmental disorders, are associated with any specific alterations in the dynamics of spindle generation, including potential subdivisions of spindles into 'slow' or 'fast' types around the SW.

## Relationship to previous findings – slow waves

We observed increased SW amplitude in 22q11.2DS and mediation of genotype differences in anxiety and ADHD symptoms by SW amplitude (and spindle-SW coupling). A previous study found delta frequency (<4 Hz) EEG activity to be reduced in ADHD patients not using psychostimulant medication (*Furrer et al., 2017*) the authors related their finding to reduced cortical grey matter, and delays in its maturation in ADHD (*Nakao et al., 2011*; *Shaw et al., 2006*; *Shaw et al., 2010*). In contrast, imaging studies have suggested increased cortical grey matter thickness in 22q11.2DS, alongside changes in corticothalamic networks (*Lin et al., 2017*; *Sønderby et al., 2022*; *Sun et al., 2020*), which may reduce across adolescence (*Schaer et al., 2009*). This could therefore explain our finding of increased SW amplitude in 22q11.2DS, and its relationship with ADHD symptoms, as it has been previously demonstrated (in adults) that greater SW amplitude is associated with greater cortical thickness (*Dubé et al., 2015*).

Anxiety and ADHD symptoms in late childhood (~age 10) are associated with subsequent psychotic symptoms in 22q11.2DS (*Chawner et al., 2019*; *Niarchou et al., 2019*), although ASD symptoms are not. Brain imaging studies have demonstrated that individuals with 22q11.2DS who developed psychotic symptoms had a trajectory of thicker frontal cortex in childhood and early adolescence, which then more rapidly thinned during adolescence, than individuals with 22q11.2DS who did not develop psychotic symptoms (*Bagautdinova et al., 2021*; *Ramanathan et al., 2017*). Therefore, increased spindle and SW amplitude in 22q11.2DS in childhood/adolescence could reflect aberrant cortical development processes which clinically associate with ADHD and/or anxiety symptoms in

this age group, but then progress to thinner frontal cortex, increased risk of psychotic disorders and potentially decreased spindle/SW density in adulthood.

## Relationship to previous findings – aperiodic signal component

We discovered increased broadband EEG power in 22q11.2DS during sleep, particularly in REM. Furthermore, the intercept of the aperiodic signal component in REM was observed to be a mediator of genotype effects on ADHD symptoms. One possibility is that the increased power is related to the increased cortical grey matter thickness observed in 22q11.2DS, as has been observed in brain imaging studies (*Lin et al., 2017*; *Sønderby et al., 2022*; *Sun et al., 2020*).

We also observed that the slope and intercept of the aperiodic part of the signal reduced with age, similar to previously reported findings in awake resting state EEG in children and adolescents (*Hill et al., 2022*), and aging adults (*Voytek et al., 2015*). The slope of the 1 /f component of the EEG has also been associated with changes in level of arousal across different sleep stages, with REM being associated with the steepest slopes (*Kozhemiako et al., 2021*; *Lendner et al., 2020*). However, we did not observe any differences between groups in 1 /f slope.

## Mechanisms of sleep EEG changes in 22q11.2DS

Our EEG findings together suggest a complex picture of sleep neurophysiology in 22q11.2DS. On the one hand, increased intercept of the aperiodic component of the signal and increased SW amplitude is associated with a younger developmental age in controls; on the other hand, higher spindle frequency and higher spindle-SW coupling is associated with an older developmental age. Furthermore, we found partial mediation of genotype effects on anxiety, ADHD and ASD symptoms by several EEG measures, in addition to opposing relationships between EEG measures and memory task performance in 22q11.2DS, again suggesting a complex relationship between sleep physiology and cognition in carriers of this genotype.

Although the physiological bases of 22q11.2DS-associated changes in sleep architecture are unknown, genes in the deleted region of chromosome 22 have been implicated in sleep regulation, potentially via a role in sleep promoting GABA-ergic signaling (*Maurer et al., 2007*). GABA signaling is also integral to the mechanisms of spindle and slow wave oscillations (*Feld et al., 2013*; *Feld and Born, 2020*; *Ulrich et al., 2018*). Although a study using magnetic resonance spectroscopy did not demonstrate gross changes in GABA levels in the anterior cingulate cortex in 22q11.2DS (*Vingerhoets et al., 2020*), a mouse model of 22q11.2DS does harbor a reduction in GABA-ergic parvalbumin containing cortical interneurons in 22q11.2DS (*Al-Absi et al., 2020*). Therefore, a potential mechanism of altered sleep and sleep-associated EEG oscillations may involve neurodevelopmental changes to cortical structure and GABAergic signaling in cortical inhibitory networks in 22q11.2DS.

## Limitations and future directions

In this study we made a single overnight EEG recording. Although there were no differences between groups in terms of total sleep time, or awakenings overnight – indicating minimal disruption by the EEG recording setup – it is possible that the recording protocol caused undetected effects that differed between groups, contributing to our EEG findings. A future study which included a baseline night where participants become familiar with the recording equipment would help to address this possibility.

We used interaction and mediation analyses to infer associations between genotype, psychiatric measures or FSIQ, memory task performance, and a wide range of sleep EEG measures. These initial findings should be replicated in a larger sample to confirm sleep EEG measures as intermediate phenotypes that predict behaviour and cognition. Here we need to emphasise that the discovered correlations between sleep EEG measures and memory performance in the morning do not directly relate to processes associated with sleep dependent memory consolidation.

Sleep architecture is heterogenous in young people with ADHD and ASD, as it is in adult schizophrenia patients (*Chouinard et al., 2004*; *Cohrs, 2008*); it therefore may be unlikely that sleep macrostructure alone (i.e. percentages of different sleep stages, sleep efficiency etc.) will prove a useful biomarker or prognostic indicator of later neurodevelopmental diagnoses in 22q11.2DS. However, our results suggest that the use of quantitative measurements of sleep microstructure, such as of spindles, SWs and their coupling could be mediators of genotype effects on psychiatric symptoms

and therefore be useful as biomarkers of neurodevelopmental disorders in future studies (*Manoach and Stickgold, 2019*).

As expected, age had a large influence on EEG properties in our between-subject, cross-sectional study (*Hahn et al., 2020*; *Markovic et al., 2020*; *Purcell et al., 2017*). It has previously been demonstrated that psychopathology changes with age in 22q11.2DS, including that ADHD symptoms decline with age (*Chawner et al., 2019*). Therefore, a longitudinal cohort study of sleep EEG biomarkers in 22q11.2DS from childhood and adolescence into adulthood is an important extension of the present study, to elucidate developmental trajectories, as has been achieved with brain imaging (*Bagautdinova et al., 2021*; *Ramanathan et al., 2017*). Further, a retrospective cohort study of EEGs for those with 22q11.DS who go on to develop schizophrenia-spectrum disorders could dissociate which EEG features relate to the development of psychosis.

## Conclusion

In conclusion, in this study quantifying sleep neurophysiology in 22q11.2DS, we highlight differences that could serve as potential biomarkers for 22q11.2DS-associated neurodevelopmental syndromes. Future longitudinal studies should clarify the relationship between psychiatric symptoms, sleep EEG measures, and development in 22q11.2DS, with a view to establishing mechanistic biomarkers of circuit dysfunction that may inform patient stratification and treatment.

## Materials and methods

### Participants and study recruitment

Participants were recruited as part of the previously described, ongoing Experiences of Children with cOpy number variants (ECHO) study (*Moulding et al., 2020*). Where available, a sibling (n=17) without the deletion closest in age to the participant with 22q11.2DS (n=28) was invited to participate as a control. As this study was an exploratory cross-sectional study of a rare neurodevelopmental copy number variant, our sample size was taken as the maximum number of participants who agreed to have an EEG recording.

The presence or absence of the deletion was confirmed by a Medical Genetics laboratory and/ or microarray analysis in the MRC Centre for Neuropsychiatric Genetics and Genomics laboratory at Cardiff University.

Prior to recruitment, primary carers consented for all participants and additional consent was obtained from participants aged ⩾16 years with capacity. The protocols used in this study were approved by the NHS Southeast Wales Research Ethics Committee.

Age and sex characteristics of the study sample are shown in *Table 1*. Of participants with 22q.11.2DS, four were prescribed melatonin, one was prescribed methylphenidate (Medikinet) for ADHD and one was prescribed sertraline for 'mood'. No controls were prescribed psychiatric medication. No study participant reported a diagnosis of epilepsy or seizure disorder.

All data were collected during study team visits to participants' family home. Data collection, including EEG recordings from sets of siblings were carried out on the same visit, which were typically on weekends or school holidays.

### Psychiatric characteristics and IQ

Psychopathology and subjective sleep quality was measured by the research diagnostic Child and Adolescent Psychiatric Assessment (CAPA) interview (*Angold et al., 1995*) with either the participant or primary carer. Interviews were carried out during the same visit as EEG recordings. Participants were also screened for Autism-Spectrum Disorder (ASD) symptoms using the Social Communication Questionnaire [SCQ, (*Rutter et al., 2003*)], completed by the primary carer. Full-Scale IQ FSIQ was measured using the Wechsler Abbreviated Scale of Intelligence (*Wechsler, 1999*), as the combination of all subscores.

### Sleep-dependent memory consolidation task

The effect of sleep on participants' memory performance was evaluated using a 2D object location task (*Wilhelm et al., 2008*) implemented in *E-Prime*. Participants completed a learning and test session the evening before the EEG recording, and a recall session the next morning. In the task, a

5x6 grid of covered square 'cards' was presented on a laptop screen. During learning, successive pairs (n=15) of cards were revealed for 3 s, showing matching images of everyday objects and animals. During recall, one of each pair was uncovered, and subjects were required to select the covered location of the matching pair.

In the evening, learning and recall sessions alternated until participants reached a performance criterion of 30% accuracy. The next morning a test session was carried out with a single recall session.

## Polysomnography data acquisition

High-density EEG and video recordings were acquired with a 60 channel Geodesic Net (Electrical Geodesics, Inc Eugene, Oregon, USA) and a BE Plus LTM amplifier running the Galileo acquisition software suite (EBNeuro S.p.A, Florence, Italy). Additional polysomnography channels including EOG, EMG, ECG, respiratory inductance plethysmography, pulse oximetry and nasal airflow were recorded with an Embla Titanium ambulatory amplifier. PSG signals were acquired at 512 Hz sampling rate with a 0.1 Hz high pass filter. The online references was electrode Cz.

On recording visits, a member of the study team came to the participant's home, set up the EEG and PSG recording systems and left; participants went to be at their normal bedtime, slept in their own beds, and as the system was ambulatory, were able to move freely during recordings for example to use the bathroom overnight. The experimenter returned the following morning to end the recordings, collect equipment and carry out the morning memory task test session.

## Sleep scoring

Sleep scoring was performed by an experienced scorer on a standard PSG montage (6 EEG + all PSG channels) according to Academy of American Sleep Medicine criteria. Artefact and Wake epochs of EEG were discarded from further analysis. Sleep architecture was quantified using standard derived variables: total sleep time, sleep efficiency, latencies to N1 and REM sleep and proportion of time spent in N1, N2, N3, and REM sleep.

## EEG data analysis

All pre-processing, spectral analysis and event detection algorithms employ methods validated in previously published sleep EEG studies, using the same MATLAB code where available.

## Pre-processing

Following acquisition and prior to analysis, EEG data was pre-processed using the following steps: EEG.*edf* files were imported into MATLAB (Mathworks, Nantick, MA, USA) using the *EEGLAB* toolbox (*Delorme and Makeig, 2004*). Next, signals were downsampled to 128 Hz and processed: detrended with a high pass filter (cut off 0.25 Hz), 50 Hz line noise removed using the *EEGLAB PREP* plugin (*Bigdely-Shamlo et al., 2015*), artefacts were removed with the Artifact Subspace Reconstruction method (*Mullen et al., 2015*) implemented in the clean_rawdata EEGLAB plugin and re-referenced to a common average. Two additional automated artefact removal steps were then applied. Firstly, full 60-channel recordings were decomposed with independent components analysis (ICA) using the AMICA EEGLAB plugin (*Delorme et al., 2012*) and non-brain components were removed using the ICLabel plugin, including ECG, EMG, and EOG artifacts (*Pion-Tonachini et al., 2019*).

Second, we applied an automated iterative epoch-level artefact removal process to N2 and N3 epochs (pooled together) using two sets of criteria, similar to that described by *Purcell et al., 2017*: firstly, we applied the method described by *Buckelmüller et al., 2006*, removing epochs where the beta (16–25 Hz, 2 SD) or delta (1–4 Hz, 2.5 SD) power exceeded a threshold of 2 or 2.5 SD relative to the flanking 14 epochs (7 before and 7 after). The resulting set of epochs were then further filtered based on whether an epoch had >5% clipped signals (e.g. >5% of values in the epoch equal to the minimum or maximum possible value) and then a three-cycle iterative process of removing epochs based on their having a signal RMS or score on the first3 parameters (*Hjorth, 1970*) that exceeded 2 SD of the whole-signal SD. These processes appeared to remove epochs randomly across the night, although may have removed N3 epochs more than N2. There were no group differences in the proportion of N2 or N3 sleep removed from each recording (mixed models fit to proportion of epochs in N2 and N3 separately, with group as independent variable and gender and age as covariates, both p>0.05). This process was applied to each channel independently as this study did not investigate

any cross-channel EEG measures. Sleep montages were then prepared for sleep scoring, and thirty second epochs containing artefacts were flagged for removal after manual review.

## Time-frequency Analysis

We calculated whole-night spectrograms using the multitaper method (*Bokil et al., 2010*) with a 30 second window advanced in steps of 10 seconds and a bandwidth of 1 Hz. The EEG power spectral density (PSD) was calculated for frequencies between 0.25 and 20 Hz using Welch's periodogram (MATLAB function *pwelch* with a 4 second Hanning window advanced in 2 second steps, then averaged over time to give one value per frequency per epoch) and converted to decibels ($10*\log_{10}$ (microvolts$^2$)). We then repeated this analysis with EEG signals after z-scoring in signals in the time domain (subtracting the overall signal mean and dividing by the signal standard deviation).

Next we used the Irregular Resampling Auto-Spectral Analysis (IRASA) method (*Wen and Liu, 2016*) to decompose EEG signals into oscillatory and aperiodic (also known as 1 /f or fractal) components, using the MATLAB code published by the method's authors. In brief, this method consists of sequentially resampling time domain signals to odd numbered sampling rates and calculating the PSD for this set of stretched or compressed data. The sum of these resampled datasets cancels out oscillatory activity, but the aperiodic component is retained. The oscillatory component of the signal can then be recovered by subtracting the aperiodic component from the full PSD in the frequency domain. We calculated differences in PSD between groups through linear mixed models fit at each frequency, and corrected p-values for multiple comparisons using cluster-based correction [where adjacent frequencies were considered to be neighbors (*Maris and Oostenveld, 2007*), with 500 permutation iterations].

From the oscillatory component of the signal, we calculated average power in slow delta (<1.5 Hz) and sigma bands (10–16 Hz), and the sigma-band frequency with maximum power, averaging over all epochs in N2 and N3 sleep separately for each subject. From the aperiodic component of the signal, we calculated two measures – a slope and an intercept measure, from fitting an exponential of $y=e^{-slope}+$intercept to the aperiodic data in the frequency domain. We calculated these measures from all N2, N3 and REM epochs separately for all individuals. This gave a total of 12 spectral measures per electrode per subject (*Table 7*).

## Spindle detection

Sleep spindles were automatically detected from artefact-free epochs of N2 and N3 sleep EEG data using a relative-threshold detector based on previously reported methods using the continuous wavelet transform (*Djonlagic et al., 2021*; *Purcell et al., 2017*). To enhance spindle detection signal-noise ratio a Complex-Frequency B-Spline wavelet was used in the place of the more typical Complex Morlet wavelet (*Bandarabadi et al., 2020*).

We determined the wavelet frequency to use for spindle detection from the peak sigma frequency calculated using the IRASA method, for each individual. As we observed almost all participants to have unimodal distributions of sigma power (*Figure 1—figure supplement 1*), and as has been previously suggested in a study of similarly aged participants (*Hahn et al., 2020*), we did not differentiate between 'fast' and 'slow' spindles. Therefore, for spindle detection, we used a single wavelet with an individualized centre frequency, a bandwidth parameter of 2, and an order parameter of 2.

Putative spindle cores were identified from the magnitude of the continuous wavelet transform of the EEG signal (smoothed with a 0.1 s moving average), with a main threshold of 3 x the median (calculated over the whole signal) and a secondary threshold of 1.5 x the median. We took putative spindles to be crossings of the main threshold flanked with secondary threshold crossings with a minimum event duration of 0.5 s and a maximum duration of 3 s. Further, putative spindles had to be separated by at least 0.5 s; events closer than this were merged unless their combined duration exceeded 3 s.

Putative spindles were further selected based on a quality metric where the power increase in the sigma band (10–16 Hz) during the putative spindle event (calculated as the FFT of the signal, relative to the whole-night baseline PSD) had to exceed the average increase in the delta, theta and beta bands during the same period, relative to their whole-night baselines.

From each putative spindle we extracted the amplitude (maximum peak-to-trough voltage difference in the sigma filtered-EEG, filtered using a least-squares linear-phase FIR filter using MATLAB's

*firls* command with 10–16 Hz passband filter, an order of 960, and transition frequency width of 0.5 Hz) and frequency (reciprocal of the mean time difference between positive voltage peaks within the spindle). We also calculated the average density of spindles over the whole duration of all epochs investigated in each participant, in events per minute, giving a total of three spindle-related measures per electrode per subject (*Table 7*).

## Slow-wave detection

Slow waves were detected from epochs of N2 and N3 sleep using a previously validated method (*Djonlagic et al., 2021*) as follows: first the EEG signal was band-pass filtered between 0.25 and 4 Hz using a Hamming windowed sinc FIR filter (*pop_eegfiltnew* from the EEGLAB toolbox), Next, negative half-waves were detected from positive-to-negative zero crossings and selected as putative SWs if: the putative SW had an amplitude greater 2 x the signal median for all negative half-waves, a minimum length of 0.5 s and a maximum length of 2 s. These were liberal criteria which detected large numbers of negative half waves; this approach was chosen as we wished to investigate SW-spindle interactions, and therefore wished to maximise our sample of putative SWs. Furthermore, as it has been observed that the overall power of the EEG signal decreases from childhood to adolescence (*Hahn et al., 2020*), the use of an absolute threshold for SW detection could introduce bias in detections based on participant age. We therefore considered a relative threshold most appropriate for our dataset.

From each SW we extracted the amplitude (as the peak negative deflection), the duration (the time between the initial positive-to-negative zero crossing to the negative-to-positive zero crossing), and from the total set of SWs calculated the average SW density in events per minute, giving a total of 3 SW-related measures per electrode per subject (*Table 7*).

To explore the polarity of EEG signals across the scalp at the time of SWs detected on individual channels, we made topoplots of the average EEG voltage at all electrodes at the time of detected SW troughs at a range of seed electrodes which were selected for being placed evenly across the scalp.

## Spindle–SW coupling

Spindle - SW coupling was measured using three metrics. First, we calculated the simple proportion of detected spindles whose peaks overlapped with any detected SWs, where overlap was defined as spindles whose peak sample fell within a window of +/- 1.5 s of the negative peak of a detected SW. Second, we calculated the mean resultant length (MRL) vector of the phase of the slow oscillation at the time of peak spindle amplitude, for spindles which overlapped an SW. The MRL metric was calculated as follows: the phase angle of the filtered slow oscillation signal was calculated using the Hilbert transform (where 0 degrees was the first positive-to-negative crossing of the SW). The phase value at the index of each spindle peak was taken, and the overall MRL for that signal was calculated as *mrl = abs(mean(exp(1i\*phase)))*, where phase is a vector of all spindle peak phase values in a recording. Third, we took the mean angle of the SW phase at the peak of all SW-overlapping spindles where *angle = angle(mean(exp(1i\*phase)))*. For the overlap and MRL measures, we converted the raw measure to a z-score relative to a resampling distribution calculated by randomly shuffling each spindle peak either within its local 30-s epoch (overlap measure) or shuffling only within the overlapping detected SW (MRL measure). This resampling procedure was repeated 1000 times to create a null distribution from which the mean and standard deviation was calculated for deriving the z-score for each signal. We then calculated these three measures for each electrode, for each subject (*Table 6*).

As an additional analysis to explore the relationship between SWs and sigma-frequency activity, we made scalograms of SW-trough-locked EEG signals, using a set of frequencies evenly spaced between 8 and 16 Hz, and a Complex-Frequency B-Spline wavelet with bandwidth 2 and order 2. Scalograms were normalised by z scoring relative to the average signal in the window 2 – 1.5 seconds prior to SW troughs. Normalised scalograms were calculated for all SWs detected at each electrode and participant, then averaged.

## Statistical analysis

Following acquisition, preprocessing and feature identification and extraction, summary data were exported into R 4.1.0 (*R Development Core Team, 2017*) for statistical analysis.

## Psychiatric, sleep architecture, and memory data

Psychiatric and sleep architecture data were analyzed using a mixed model approach, with subject family entered into models as a random (varying) intercept, to account for the shared genetic and environmental influences within sibling pairs. Mixed models are also robust to missing data.

For memory task data, the number of learning cycles to reach the 30% performance criterion were modeled using mixed effects Cox Proportional Hazard Regression (*coxme* in the *coxme* package), with right censoring as some participants (22q11.2DS n=6, siblings n=0) completed numerous training cycles but never reached criterion before stopping the task.

Accuracy in the morning memory test was measured using a binomial model [*glmer* with *family = binomial(link = "logit")* in the *lme4 package* (**Bates, 2010**)] with the number of hits from the 15 trials as the dependent variable. In both models, genotype was the independent variable and age and sex were included as covariates, with family as a random intercept.

## High-density sleep EEG data

Our EEG dataset consisted of 21 EEG measures across 60 channels recorded in 45 participants (*Table 7*). As we used a high-density electrode array, we used multilevel generalized additive models (GAMM) applied to EEG data from all 60 electrodes in one model. The generalized additive model can model non-linear relationships, including spatial relationships, between variables using splines or other smoothers (**Wood, 2004**; **Wood, 2017**) and can, for example, be used to model complex or spatial relationships in models that also include covariates and random effects structures (**Pedersen et al., 2019**). In the case of EEG data, this allowed the estimation of a model than included the value of an EEG measure of interest (e.g. spindle density) at all 60 electrodes, with genotype, gender and age as a covariates and participant identity nested in their family identity as a random intercept, giving overall statistics for group differences.

This approach represents an extension of traditional EEG topographical plots, which present a smoothed interpolation of an EEG measure onto a 2D grid representing the head, by allowing the modelling of the relationship between multiple independent variables (and random effects) across the 2D representation of the head.

In order to fit these computationally intensive models consistently, we adopted a Bayesian approach, fitting the models using Hamiltonian Monte Carlo using the R *BRMS* package and Stan (**Bürkner, 2017**; **Carpenter et al., 2017**), with 4000 iterations (1000 warm up) on four chains, and regularizing priors [in keeping with (**McElreath, 2018**)]. All model posterior information was inspected manually, and all Gelman-Rubin statistics (measures of model convergence) were close to 1.00, which is considered the optimal value. The formula used for all models (except for angular data) was *value ~s(x, y, by = group, bs = "tp", k=20, m=2)+gender + age_eeg +group + (1|family) + (1|family:subject)*, where *value* was a placeholder for each EEG feature of interest, and *x* and *y* represented the 2D-projected x and y co-ordinate for each EEG electrode. This model fits an isotropic smooth with thin plate regression splines to the EEG measure across electrodes, with a separate smooth being estimated for 22q11.2DS and sibling controls. The model was fit was a normal distribution and an identity link function.

This approach allowed us to generate topographic plots of group differences by taking draws from the posterior fitted values of each model across a grid of spatial locations, for both 22q11.2DS and Siblings. From these posterior samples, we then calculated the genotype difference distribution, and from this, calculated the probability of direction statistic (**Makowski et al., 2019**), selecting spatial locations where the value of the statistic was less than 0.05 for further plotting.

## Spindle-SW coupling angle model

We modelled Spindle-SW Coupling angle (in radians) with a GAMM with a von Mises distribution (equivalent to a normal distribution on the circle) and a half tan link function. We modelled both the mean angle and kappa (the circular concentration parameter) to improve model fitting. The formula used for the angle model was: *overlap_angle ~0 + group + gender + age_eeg +s(x,y,by = group, bs = "tp", k=20, m=2) + (1|family) + (1|family:subject), kappa ~0 + group + gender + age_eeg + s(x,y,by = group, bs = "tp", k=20, m=2) + (1|family) + (1|family:subject)*. We plot both the mean angle for coupling across the scalp, and model estimated angle differences between groups on topoplots with a circular color scale.

### Memory task – EEG correlation models

We analyzed the correlation between Sleep EEG measures and performance in the test session of our behavioral task the next morning. For each EEG measure and EEG electrode, we fit a generalized linear mixed model, with number of hits in the morning session as dependent variable, the interaction between an EEG measure and genotype as independent variable, and gender and age as covariates, with family identity as a random intercept, and a binomial distribution with logit link function. We then took p-values for the interaction term, corrected using a cluster-correction permutation testing approach with 500 permutations (*Maris and Oostenveld, 2007*), and plotted those electrodes where the EEG measure * genotype interaction was significant (cluster-corrected p<0.05).

### EEG mediation models

We analyzed whether genotype effects on psychiatric symptoms or FSIQ were mediated by EEG measures using mediation models (*Imai et al., 2010*). Mediation analysis is a statistical technique which allows the estimation of whether the effect of one exposure (genotype) on an outcome (here a psychiatric measure or FSIQ) occurs via an effect of the exposure on a third variable (known as the mediator, here an EEG measure) or via a direct effect on the exposure. These models can be estimated by the combination of two statistical models predicting (1) the outcome from the exposure, mediator, and other covariates (including random effects) and (2) the mediator from the exposure, other covariates and random effects.

In our mediation analysis, we fit models for each pair of a psychiatric measure or IQ (ADHD, anxiety, ASD symptoms, psychotic experiences, CAPA sleep problems or FSIQ) and an EEG measure (focusing on the four measures where we observed raw group differences: REM constant, spindle amplitude, SW amplitude and spindle-SW MRL): one linear mixed model predicting the EEG measure from genotype, age and gender as covariates, and family as a random intercept; and one model predicting the psychiatric measure/IQ from genotype, EEG measure, age and gender, and family as a random intercept. The link function of the second model depended on the measure; poisson count models were used for symptom count data (ADHD, anxiety and ASD), a binomial model for psychotic experiences, or a linear model for FSIQ. These models were fit using the *lme4* package as above, then combined in a mediation analysis using the *mediate* function in the *R mediation* package (*Imai et al., 2010*). From each model we extracted the estimated direct and mediated effects, and the proportion mediated (the proportion of the total effect of genotype on the psychiatric measure/FSIQ mediated via the EEG measure) and constructed topographic plots of the proportion mediated. We derived p-values from mediation models using the cluster correction method by generating 500 shuffled datasets at each electrode, where group identity was permuted, and refit the models to each. We then plotted those electrodes where there was a cluster-corrected significant mediated effect by the given measure *and* a significant total effect on the same electrode.

## Acknowledgements

We are extremely grateful to all the families that participated in this study as well as to support charities Max Appeal, The 22Crew and Unique for their help and support. We thank the core laboratory team of the Division of Psychological Medicine and Clinical Neurosciences laboratory at Cardiff University for DNA sample management and genotyping, and the National Centre for Mental Health, a collaboration between Cardiff, Swansea and Bangor Universities, for their support. The authors also thank Dr Ines Wilhelm for kindly donating the 2D object location task implementation.

## Additional information

### Competing interests

Ullrich Bartsch, Hugh Marston: was full-time employees of Lilly UK during this study. Michael J Owen, Marianne BM van den Bree: reports a research grant from Takeda pharmaceuticals outside the scope of the current study. The other authors declare that no competing interests exist.

## Funding

| Funder | Grant reference number | Author |
|---|---|---|
| National Institute of Mental Health | NIMH 5UO1MH101724 | Marianne BM van den Bree |
| Eli Lilly and Company | Lilly Innovation Fellowship Award | Ullrich Bartsch |
| National Institute for Health and Care Research | Academic Clinical Fellowship | Nicholas A Donnelly |
| Baily Thomas Charitable Fund | 2315/1 | Marianne BM van den Bree |
| Waterloo Foundation | 918-1234 | Marianne BM van den Bree |
| Medical Research Council | MR/L011166/1 | Jeremy Hall |
| Medical Research Council | MR/T033045/1 | Jeremy Hall |
| Wellcome Trust | 100202/Z/12/Z | Jeremy Hall |
| Wellcome Trust | 202810/Z/16/Z | Matt W Jones |
| Medical Research Council | 1644194 | Christopher Eaton |
| Medical Research Council | MR/K501347/1 | Hayley A Moulding |

For the purpose of Open Access, the authors have applied a CC BY public copyright license to any Author Accepted Manuscript version arising from this submission. The funders had no role in study design, data collection and interpretation, or the decision to submit the work for publication.

## Author contributions

Nicholas A Donnelly, Resources, Data curation, Software, Formal analysis, Visualization, Methodology, Writing – original draft, Writing – review and editing; Ullrich Bartsch, Conceptualization, Resources, Data curation, Software, Formal analysis, Supervision, Funding acquisition, Investigation, Project administration, Writing – review and editing; Hayley A Moulding, Data curation, Formal analysis, Validation, Investigation, Project administration; Christopher Eaton, Data curation, Validation, Investigation, Project administration; Hugh Marston, Conceptualization, Supervision, Funding acquisition; Jessica H Hall, Data curation, Project administration, Writing – review and editing; Jeremy Hall, Michael J Owen, Marianne BM van den Bree, Matt W Jones, Conceptualization, Supervision, Funding acquisition, Writing – review and editing

## Author ORCIDs

Nicholas A Donnelly ⬤ http://orcid.org/0000-0003-2234-8545
Ullrich Bartsch ⬤ http://orcid.org/0000-0002-2213-8989
Christopher Eaton ⬤ http://orcid.org/0000-0001-6739-1999

## Ethics

Human subjects: The consent process and ethical approval is described in detail in the manuscript (page 32).

## Decision letter and Author response

Decision letter https://doi.org/10.7554/eLife.75482.sa1
Author response https://doi.org/10.7554/eLife.75482.sa2

# Additional files

## Supplementary files

• Transparent reporting form

• Source data 1. The zip file "eLife Submission Data.zip" contains data files which are used to produce the analysis and figures presented in the manuscript. Note that data files are in the .rds file format, which can be opened using the free open source R statistic programming language. Source data 1: eLife Submission Data.zip sleep_study_beh_psych_demographic_data.rds: Contains

data for participant performance on the memory task, psychiatric and cognitive assessments and demographic data sleep_study_eeg_coupling_example.rds: Contains EEG data of an example spindle-SW interaction event, used to construct Figure 4A sleep_study_eeg_locations.rds: Contains the standardized locations of the EEG electrodes used throughout the study sleep_study_eeg_ neighbours.rds: Contains the neighboring electrode for all 60 electrodes in the recording system used in our study, used in the cluster correction statistics utilized in Figure 5B and Figure 6B sleep_ study_eeg_spectrogram_example.rds: Contains example whole night spectrograms for a sibling pair, with hypnogram and spindle and SW detection counts over the night, used in Figure 3A-B sleep_study_eeg_summary_data.rds: Contains summary EEG measures (one value per measure per electrode per participant) for the measures in Table 7. Data are in their own standard units. sleep_study_eeg_summary_data_z.rds: Contains summary EEG measures (one value per measure per electrode per participant) for the measures in Table 7. Data are z scored. sleep_study_epoch_ removal.rds: Contains data on the number of epochs removed by our artefact removal process (See Methods and Materials/EEG data analysis/Pre-processing) sleep_study_example_so_waveforms_ bootci.rds: Contains example slow wave waveforms and bootstrapped 95% confidence intervals, used in Figure 3D sleep_study_example_spindle_waveforms_bootci.rds: Contains example spindle waveforms and bootstrapped 95% confidence intervals, used in Figure 3C sleep_study_interaction_ clusters.rds: Contains the electrode locations and statistics for clusters of significant group*EEG measure interactions, used in Figure 5B sleep_study_interaction_models.rds: Contains fitted mixed models for group*EEG measure interactions, used in Figure 5B sleep_study_mediation_ plot.rds: Contains cluster-corrected mediation models, used in Figure 6B sleep_study_psd_bootci. rds: Contains power spectral density plot data, with bootstrapped 95% confidence intervals, used in Figure 2A-D sleep_study_psd_clusters.rds: Contains the location of clusters of frequencies with significant group differences, used in Figure 2A-D sleep_study_psd_data.rds: Contains PSD data used to calculate group differences and for cluster correction, to derive the data in sleep_study_ psd_bootci.rds and sleep_study_psd_clusters.rds sleep_study_topoplot_angle_data.rds: Contains posterior samples from the GAMM model fit to the coupling angle data. Used to construct the topoplot found in Figure 4D sleep_study_topoplot_posterior_data.rds: Contains posterior samples from the GAMM models used to construct group difference topoplots for the EEG measures in Table 7 (except angular data, which is found in) sleep_study_topoplot_angle_data.rds. Used to construct the topoplots found in Figure 2E-G, Figure 3E-F and Figure 4C

• Source code 1. The .zip file "eLife Submission Scripts.zip" contains R scripts that produce the main figures and analysis. Scripts beginning with the prefix "Analysis" run computations which produce data used in tables and figures (and may run slowly); scripts beginning with the prefix "Figure" produce the figures themselves. The following files are included: Analysis-Artefact-Removal.R: Code to analyze the effect of artefact removal (See Methods and Materials/EEG data analysis/ Pre-processing) Analysis-Common-Utilities.R: Code that contains common variables and functions used in the paper Analysis-fitGAMMs.R: Code that fits the Bayesian GAMMs and extracts posterior samples which produce the topoplots of group differences in Figure 2E-G, Figure 3E-F and Figure 4C Analysis-fitMediationModels.R: Code that fits the mediation models and performs cluster-based correction for multiple comparisons, producing data used to plot Figure 6B Analysis-Interaction-Clusters.R: Code that fits the EEG*genotype interaction models and performs cluster-based correction for multiple comparisons, producing data used to plot Figure 5B Analysis-PSD-Clusters.R: Code that fits models to the PSD data across multiple frequencies, and performs cluster-based correction for multiple comparisons, producing the data used to plot Figure 2A-D Analysis-Tables-1-5.R: Code that produces Tables 1-5, including the statistical tests presented in those tables Figure-1-Revised.R: Code to produce the elements of Figure 1 and associated figure supplements Figure-2-Revised.R: Code to produce the elements of Figure 2 and associated figure supplements Figure-3-Revised.R: Code to produce the elements of Figure 3 and associated figure supplements Figure-4-Revised.R: Code to produce the elements of Figure 4 and associated figure supplements Figure-5-Revised.R R: Code to produce the elements of Figure 5 Figure-6-Revised.R R: Code to produce the elements of Figure 6 and Table 7

## Data availability

All data generated or analysed during this study are included in the manuscript and supporting file; Source Data files have been provided as a .zip file. Extensive additional information collected as part of the ongoing IMAGINE-ID study, of which the ECHO study forms part, can be obtained via https:// imagine-id.org/healthcare-professionals/datasharing/.

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
