## [Editor Report]

The authors quantified sleep oscillations and their coordination in young people with 22q11.2 Deletion Syndrome and their siblings. This was done to identify potential biomarkers of later neurodevelopmental diagnoses in 22q11.2 Deletion Syndrome. The core findings based on solid data demonstrate that sleep rhythms in 22q11.2DS are altered in comparison to the control group, as is their relationship with the behavioral expressions of memory consolidation. These are important findings as they directly provide a link between genes and sleep rhythms and memory consolidation.

---

## [Decision Letter]

**Decision letter after peer review:**

Thank you for submitting your article "NREM sleep EEG in young people with 22q11.2 deletion syndrome: slow-waves, spindles and interrelationships with memory and neurodevelopmental symptoms" for consideration by *eLife*. Your article has been reviewed by 3 peer reviewers, and the evaluation has been overseen by a Reviewing Editor and Laura Colgin as the Senior Editor. The following individuals involved in review of your submission have agreed to reveal their identity: Malgorzata Wislowska (Reviewer #1); Leila Tarokh (Reviewer #2).

Essential revisions:

In particular, several of the referees have raised concerns about whether the conclusions are supported by the data. Please revise the paper according to the concerns raised below with a specific focus on revising the conclusions in the light of experimental results.

*Reviewer #1 (Recommendations for the authors):*

Limitations of the study:

1. Detected slow waves

Many of the detected slow waves has amplitude below 75mV (Figure 4B). Furthermore, the highest density of slow waves in 22q group is over parieto-occipital scalp sites (Figure 4C). This makes me wonder whether the authors didn't detect events that aren't slow waves. In the methods section, the authors claim to use liberal criteria to increase number of detections. While I can understand this motivation, especially for the coupling analysis, the authors could consider re-running the analysis using standard detection criteria, to at least confirm their findings regarding slow-wave events. Alternatively, the authors could restrict their algorithm to detect the slow waves only in the signal from Fz electrode. I would be surprised if a full-night sleep data didn't have enough "regular" slow-wave events for the planned analysis.

2. State vs trait effect of memory

Figure 6: The authors reported significant relationship between success in a memory task morning recall, and sleep oscillatory features, like sleep spindles. It is however not easy to disentangle whether the reported effect is indeed related to the memory consolidation process, or to general cognitive capacities. For example, participants with higher IQ might have in general higher spindle amplitude and perform better on the memory task. It is of course reassuring, that there was no relationship between memory task performance and FSQI measure. Did the authors use for this analysis general score, or only matrix reasoning score? To zoom into memory consolidation processes, the authors could use a measure of evening-to-morning change in task performance.

Furthermore, additional mediation analysis of genotype of memory performance (preferably memory consolidation) by EEG sleep features could shed additional light on this intricated puzzle.

3. Lack of screening night

As far as I understand, sleep PSG data was recorded only once from each subject. Even though the data was acquired at participants' own homes, sleeping with an EEG cap can easily lead to disturbances of sleep architecture and sleep quality. It cannot be excluded, that there was a group difference in that respect, especially when one group of participants is clinical. In that respect, lack of an adaptation recording is a limitation of the current study and should be discussed.

Please report what was the order of data acquisition for two siblings. Was data from the siblings acquired on the same night? If not, in which order was data acquired? If the sibling with the gene mutation was always recorded first, that could induce higher stress (of unknown) in this group of participants, potentially contributing to the observed between-group differences.

4. Comparisons to the literature on adult schizophrenic patients

For the large part of the discussion, the study results are contrasted with literature on adult schizophrenic patients (e.g. line 400, line 411). However, only part (<41%) of the 22q11.2DS children are expected to develop psychotic disorders (line 74), and those people are also in risk of several other neuropsychiatric conditions (line 71). Furthermore, developmental changes are still taking place in this population (line 392), rendering additional portion of variance. Therefore, I don't find this comparison to adult schizophrenic population appropriate, and certainly not as the central point of the study.

I suggest shifting the focus towards disentangling complex relationship between the recorded parameters (gene mutation, sleep features, neuropsychiatric symptoms, overnight memory consolidation), which I find to be the most interesting and auspicious, and yet very little understandable after reading the manuscript. It is fascinating that "genes in the deleted region of chromosome 22 have been implicated in sleep regulation" (line 380), and that sleep plays a crucial role in memory consolidation, and that the previous research revealed sleep problems in 22q11.2DS population, and that neuropsychiatric symptoms are associated with sleep alterations.

5. Logic of the arguments in the Discussion section

– Line 395: or because the recorded subject will never develop schizophrenia; therefore, the conclusion in line 398 is too-far reaching.

– Line 400: the authors mention schizophrenia, then developmental changes, and then make conclusions about cortical excitability. I don't see how these pieces of information are related to each other.

– Line 405: I find these speculations to be too-far reaching. There is no evidence in the data for increased cortical excitability. The increased amplitude might as well come from increased synchronization of neural activity. I also don't understand the phrase "persistence of cortical Up-states". How this further relates to timing of coupling between spindles of slow-waves is also not clear to me.

– Line 408: I don't understand what the authors would like to test with transcranial and auditory stimulation.

– Line 416: Why does increased spindle and slow-wave amplitude indicate reduced efficiency of information processing? I find this interpretation of the memory task results to be very speculative. There are also no references in the text to support such claims.

– Line 432: Please provide references for the claim that cortical thickness leads to increased amplitude of neural oscillations.

– Line 440: Why do the authors claim that ADHD symptoms are associated with increased spindle and slow wave amplitude?

– First authors claim that "…it therefore appears unlikely that sleep macrostructure will prove a useful biomarker or prognostic indicator of later neurodevelopmental diagnoses in 22q11.2DS" (line 385) but claim something opposite in line 453.

Suggested improvements:

1. Sigma power vs sleep spindles

Page 13: Firstly, the authors analyzed spectral power, and found a between-group difference in sigma frequency power. Thereafter, the authors analyzed properties of detected sleep spindles, and showed that the only difference between the groups is in spindle power. There is a high chance that these two results reflect the same phenomenon, and are therefore redundant. Even the topo-plots for sigma power and spindle amplitude, as well as sigma frequency and spindle frequency, are very similar to each other. The authors should decide to either focus on sigma or sleep spindles.

Page 15: Are the statements made in lines 224-226 supported by the results from GAMM analysis? The authors need to include the statistics supporting the findings described in the Results sections within text, and not only in a table. This applies to the entire Results section.

2. Delta power vs slow-waves

Page 15: Similarly to the point about sleep spindles and sigma, the analysis of slow-wave properties are repeating the results from delta power analysis. The only new finding here relates to slow wave duration. As previously, I strongly suggest reporting one analysis or the other. The story is already complex by including genetic, questionnaire-based, behavioral, as well as PSG data.

If the authors decide to report sleep spindle and slow wave results, please restrict your figures to showing only the relevant findings. For example, panels A,B,C and F in Figure 2 do not provide any important information in my opinion.

3. Neuropsychiatric conditions, sleep, and 22q11.2SD

Page 6: participants with 22q11.2SD reported more sleep problems, ADHD and ASD symptoms than the control group. But a higher sleep problem was not associated with ADHD and ASD symptoms.

I suggest moving the mediation analysis (of Genotype Effects on NREM Features by Psychiatric Symptoms) to follow the previous paragraph directly. It would be also interesting to see mediation analysis of genotype on sleep problems by psychiatric symptoms. This could help explain the apparent inconsistency of the beforementioned results.

4. Sleep-spindle and slow-wave coupling

Could the authors please add a brief explanation on how MRL values should be interpreted? Does higher MRL value mean that spindles had more precise preference for occurring at a specific phase of a slow wave?

5. The manuscript needs some stylistic improvements

– explain the abbreviations on their first occurrence (e.g. ASD and FSIQ on page 6, PSD and GAMM on page 9)

– avoid repetitions if possible (sometimes the same word appears three times in one sentences)

– state explicitly which comparison do the authors refer to (e.g., line 148: "…power was significantly increased in N2 and N3 sleep in 22q11.2DS" … as compared to…?)

– avoid using phrases like "We performed first analysis…" (line 368) or "this is the first study…" (line 452), since the authors cannot know whether that's true.

– Line 433: associated with what feature of psychotic symptoms?

– Line 436: frontal cortex thicker than what?

– Please add legend for gray and blue color in every figure displaying box-plots

6. Please make sure to make factually specific claims, for example:

– Line 116: interpreting p-values above 0.05 threshold should not be interpreted as an evidence for "similarity" of two groups.

– Line 370: "…decreased [relative amount of] N1 and REM sleep…"

– Line 379: "…no [statistically significant] changes in sleep efficiency…"

7. Missing information in the Methods Section:

– What was the online EEG reference?

– Details on the study design are missing: what was the exact protocol of the experiment? When did participants go to bed? Was an experimenter staying overnight in the house during the recording? Or where parents instructed on how to deal e.g. with bathroom visits during the night? When were genetic and psychiatric tests carried out – on the same day as EEG data acquisition?

– Details about health status of participants – how many of them had which psychotic symptoms?

– Line 503: EEG is a part of PSG

– Which ICA components were removed? ECG related artefact or more?

– Why were artefacts removed only from N2 and N3, while powers spectra and sleep architecture were investigated for all sleep stages? This procedure could have artificially decrease the relative amount of N2 and N3 sleep. Furthermore, between-group differences in sleep architecture could be driven by differences in the amount of artifacts that were excluded.

– Is spindle density calculated as number of events per minute?

– Line 627: what do the authors mean with "15 EEG measures"?

– Page 642: normally distributed across what? Participants (all or within a group), or across scalp sites?

*Reviewer #3 (Recommendations for the authors):*

(1.) Line 103: remove "consolidation" from sleep dependent memory consolidation task.

(2.) Line 143: Introduce term PSD, before using abbreviation.

(3.) Line 153: The authors state that the spectral fingerprints of NREM EEG in young 22q11.2DS carriers differ from those of adult schizophrenia patients. Please briefly outline the differences (non-expert readers might be lost).

---

## [Author Response]

Reviewer #1 (Recommendations for the authors):Limitations of the study:1. Detected slow wavesMany of the detected slow waves has amplitude below 75mV (Figure 4B). Furthermore, the highest density of slow waves in 22q group is over parieto-occipital scalp sites (Figure 4C). This makes me wonder whether the authors didn't detect events that aren't slow waves. In the methods section, the authors claim to use liberal criteria to increase number of detections. While I can understand this motivation, especially for the coupling analysis, the authors could consider re-running the analysis using standard detection criteria, to at least confirm their findings regarding slow-wave events. Alternatively, the authors could restrict their algorithm to detect the slow waves only in the signal from Fz electrode. I would be surprised if a full-night sleep data didn't have enough "regular" slow-wave events for the planned analysis.

As there is no single common method for SW detection we prioritised rate of detection in order to provide a robust dataset for spindle-SW coupling analysis. We considered the use of an absolute detection threshold (e.g. – 75 microVolts) – however, because our participants were of a wide range of ages (6 to 20 years), and it is established that the absolute amplitude of the EEG decreases across childhood (e.g. Hahn et al., 2020), our view is that the use of an absolute detection threshold would potential bias the detection of slow waves by age. We have added comments on this matter to the methods section (page 37)

2. State vs trait effect of memoryFigure 6: The authors reported significant relationship between success in a memory task morning recall, and sleep oscillatory features, like sleep spindles. It is however not easy to disentangle whether the reported effect is indeed related to the memory consolidation process, or to general cognitive capacities. For example, participants with higher IQ might have in general higher spindle amplitude and perform better on the memory task. It is of course reassuring, that there was no relationship between memory task performance and FSQI measure. Did the authors use for this analysis general score, or only matrix reasoning score? To zoom into memory consolidation processes, the authors could use a measure of evening-to-morning change in task performance.Furthermore, additional mediation analysis of genotype of memory performance (preferably memory consolidation) by EEG sleep features could shed additional light on this intricated puzzle.

We used the full spectrum IQ measure, which combines all subscores in the WASI (page 33). We now present evening-to-morning change in Figure 1 and the results (page 6), and also demonstrate that FSIQ does not significantly affect performance in task acquisition or the morning test session (Table 2). Further, in light of this comment, we have revised our mediation analysis to investigate if EEG sleep features mediate genotype effects on psychiatric symptom scores and FSIQ; we did not observe any mediation by EEG markers on genotype effects on FSIQ (Figure 6).

3. Lack of screening nightAs far as I understand, sleep PSG data was recorded only once from each subject. Even though the data was acquired at participants' own homes, sleeping with an EEG cap can easily lead to disturbances of sleep architecture and sleep quality. It cannot be excluded, that there was a group difference in that respect, especially when one group of participants is clinical. In that respect, lack of an adaptation recording is a limitation of the current study and should be discussed.Please report what was the order of data acquisition for two siblings. Was data from the siblings acquired on the same night? If not, in which order was data acquired? If the sibling with the gene mutation was always recorded first, that could induce higher stress (of unknown) in this group of participants, potentially contributing to the observed between-group differences.

Your understanding is correct, and apologies for not being sufficiently clear in the previous version. We have added to the methods and discussion to address this point (page 32); we acknowledge the lack of a screening night could be a limitation of our study, although we note there were no differences in total sleep time or sleep efficiency between groups (Table 1). Data from siblings were obtained on the same nights, which we note in the methods (page 32)

4. Comparisons to the literature on adult schizophrenic patientsFor the large part of the discussion, the study results are contrasted with literature on adult schizophrenic patients (e.g. line 400, line 411). However, only part (<41%) of the 22q11.2DS children are expected to develop psychotic disorders (line 74), and those people are also in risk of several other neuropsychiatric conditions (line 71). Furthermore, developmental changes are still taking place in this population (line 392), rendering additional portion of variance. Therefore, I don't find this comparison to adult schizophrenic population appropriate, and certainly not as the central point of the study.I suggest shifting the focus towards disentangling complex relationship between the recorded parameters (gene mutation, sleep features, neuropsychiatric symptoms, overnight memory consolidation), which I find to be the most interesting and auspicious, and yet very little understandable after reading the manuscript. It is fascinating that "genes in the deleted region of chromosome 22 have been implicated in sleep regulation" (line 380), and that sleep plays a crucial role in memory consolidation, and that the previous research revealed sleep problems in 22q11.2DS population, and that neuropsychiatric symptoms are associated with sleep alterations.

We agree with the reviewer that 22q11.2 DS, although associated with psychosis, this was part of a much broader complex behavioural and psychiatric phenotype. Through multiple changes to the introduction and discussion, we believe we now better address this point, by incorporating references from other neurodevelopmental disorders relevant to 22q11.2DS (e.g. ADHD and ASD) and further discussion of putative mechanisms of sleep disruption in 22q11.2DS (discussion, page 29)

5. Logic of the arguments in the Discussion section– Line 395: or because the recorded subject will never develop schizophrenia; therefore, the conclusion in line 398 is too-far reaching.

We have extensively revised the discussion, including adding comments on the link between 22q11.2DS, and psychosis in adulthood, and how future studies could inform our understanding of this association (page 30 onward)

– Line 400: the authors mention schizophrenia, then developmental changes, and then make conclusions about cortical excitability. I don't see how these pieces of information are related to each other.

We have revised the discussion to include a specific subsection where we link our findings to the existing literature and consider putative mechanisms of 22q11.2DS related differences in EEG oscillations (page 29)

– Line 405: I find these speculations to be too-far reaching. There is no evidence in the data for increased cortical excitability. The increased amplitude might as well come from increased synchronization of neural activity. I also don't understand the phrase "persistence of cortical Up-states". How this further relates to timing of coupling between spindles of slow-waves is also not clear to me.

We have revised the discussion to include a specific subsection where we link our findings to the existing literature and consider putative mechanisms of 22q11.2DS related differences in EEG oscillations (page 29)

– Line 408: I don't understand what the authors would like to test with transcranial and auditory stimulation.

We have removed this section of the discussion as, on reflection, it did not contribute to overall argument we advance in the discussion

– Line 416: Why does increased spindle and slow-wave amplitude indicate reduced efficiency of information processing? I find this interpretation of the memory task results to be very speculative. There are also no references in the text to support such claims.

We have reviewed this point, and removed it from the discussion

– Line 432: Please provide references for the claim that cortical thickness leads to increased amplitude of neural oscillations.

References have been added to the discussion (page 29)

– Line 440: Why do the authors claim that ADHD symptoms are associated with increased spindle and slow wave amplitude?

We made this claim on the basis of the findings of our mediation analysis, which indicated that genotype effects on slow wave amplitude were mediated by ADHD symptoms (Figure 6). Following revision of the mediation analysis in line with reviewer comments as described above, we hope this point is better made; that we have statistical evidence that EEG measures are partial mediators of the effect of 22q genotype on ADHD and anxiety symptoms.

– First authors claim that "…it therefore appears unlikely that sleep macrostructure will prove a useful biomarker or prognostic indicator of later neurodevelopmental diagnoses in 22q11.2DS" (line 385) but claim something opposite in line 453.

Our tentative conclusion was that there might be a distinction in the usefulness of coarse measures like %N2 sleep and finer grained sleep microstructure such as event characteristics; we have expanded this point in the discussion (page 31)

Suggested improvements:1. Sigma power vs sleep spindlesPage 13: Firstly, the authors analyzed spectral power, and found a between-group difference in sigma frequency power. Thereafter, the authors analyzed properties of detected sleep spindles, and showed that the only difference between the groups is in spindle power. There is a high chance that these two results reflect the same phenomenon, and are therefore redundant. Even the topo-plots for sigma power and spindle amplitude, as well as sigma frequency and spindle frequency, are very similar to each other. The authors should decide to either focus on sigma or sleep spindles.

We chose to apply a systematic series of analyses to the sleep EEG data, as there have not been any previous comparisons of EEG spectral features, or EEG events in 22q11.2DS. We have further expanded on this approach, incorporating suggestions from other reviewers, to investigate the separate oscillatory and 1/f components of the PSD. We believe these analyses are both intrinsically important, and also provide important information to inform our spindle and SW-based analyses (for example using the peak sigma frequency to individualise spindle detection algorithms). We have expanded on the motivation for our approach in the results and methods sections (e.g. page 13, 36)

Page 15: Are the statements made in lines 224-226 supported by the results from GAMM analysis? The authors need to include the statistics supporting the findings described in the Results sections within text, and not only in a table. This applies to the entire Results section.

We now include relevant statistics directly in the text throughout the Results section, as suggested

2. Delta power vs slow-wavesPage 15: Similarly to the point about sleep spindles and sigma, the analysis of slow-wave properties are repeating the results from delta power analysis. The only new finding here relates to slow wave duration. As previously, I strongly suggest reporting one analysis or the other. The story is already complex by including genetic, questionnaire-based, behavioral, as well as PSG data.If the authors decide to report sleep spindle and slow wave results, please restrict your figures to showing only the relevant findings. For example, panels A,B,C and F in Figure 2 do not provide any important information in my opinion.

We have condensed our figures considerably, including making use of figure supplements to make our figures and Results sections clearer for the reader. As above, however, we believe there are important motivations for looking at PSD data before events, and carrying forward PSD-derived measures for further analysis e.g. mediation where appropriate

3. Neuropsychiatric conditions, sleep, and 22q11.2SDPage 6: participants with 22q11.2SD reported more sleep problems, ADHD and ASD symptoms than the control group. But a higher sleep problem was not associated with ADHD and ASD symptoms.I suggest moving the mediation analysis (of Genotype Effects on NREM Features by Psychiatric Symptoms) to follow the previous paragraph directly.

We have revised the structure of the Results section to place all psychiatric and behavioural data first (results, page 6), then proceed through analysis of EEG measures, before presenting analyses which combine behavioural/psychiatric measures with EEG measures. We believe this allows readers to first understand the behavioural phenotype of 22q11.2DS, then the EEG phenotype, before asking whether there are links between specific EEG measures and specific parts of the behavioural phenotype.

It would be also interesting to see mediation analysis of genotype on sleep problems by psychiatric symptoms. This could help explain the apparent inconsistency of the beforementioned results.

The relationship between sleep problems and psychiatric symptoms in 22q11.2DS, in a larger cohort, is investigated in our earlier manuscript, Moulding et al., 2020, referenced in our study, and we would invite the interested reader to peruse this.

4. Sleep-spindle and slow-wave couplingCould the authors please add a brief explanation on how MRL values should be interpreted? Does higher MRL value mean that spindles had more precise preference for occurring at a specific phase of a slow wave?

Yes, this is correct interpretation of the MRL measure. We have added an expanded explanation of this measure to the results (page 20)

5. The manuscript needs some stylistic improvements– explain the abbreviations on their first occurrence (e.g. ASD and FSIQ on page 6, PSD and GAMM on page 9)

We have reviewed the manuscript to address this

– avoid repetitions if possible (sometimes the same word appears three times in one sentences)

We have reviewed the manuscript for excessive repetition and curtailed this

– state explicitly which comparison do the authors refer to (e.g., line 148: "…power was significantly increased in N2 and N3 sleep in 22q11.2DS" … as compared to…?)

We have reviewed the manuscript to clarify the comparisons made in each section of the results

– avoid using phrases like "We performed first analysis…" (line 368) or "this is the first study…" (line 452), since the authors cannot know whether that's true.

We have removed this claim.

– Line 433: associated with what feature of psychotic symptoms?

We have added a table of the specific psychotic symptoms evinced by each participant to address this point (supplementary table 1)

– Line 436: frontal cortex thicker than what?

Based on the published literature, thicker frontal cortex in individuals with 22q11.2DS who develop psychosis has been observed relative to individuals with 22q11.2DS who do not develop psychosis, we have added this clarification (page 29)

– Please add legend for gray and blue color in every figure displaying box-plots

We have added legend to the figures where appropriate

6. Please make sure to make factually specific claims, for example:– Line 116: interpreting p-values above 0.05 threshold should not be interpreted as an evidence for "similarity" of two groups.

We have altered the results to clarify this (page 6)

– Line 370: "…decreased [relative amount of] N1 and REM sleep…"

We have altered the results to clarify this (page 6)

– Line 379: "…no [statistically significant] changes in sleep efficiency…"

We have altered the results to clarify this (page 6)

7. Missing information in the Methods Section:– What was the online EEG reference?

We had added this to the methods (page 33 – it was Cz)

– Details on the study design are missing: what was the exact protocol of the experiment? When did participants go to bed? Was an experimenter staying overnight in the house during the recording? Or where parents instructed on how to deal e.g. with bathroom visits during the night? When were genetic and psychiatric tests carried out – on the same day as EEG data acquisition?

we have added to the methods section to address these points (page 33). The experimenter did not stay overnight, the participants went to bed at their normal bedtime, and given instructions on how manage trips to the bathroom, psychiatric tests were carried out on the same visit as the EEG recording

– Details about health status of participants – how many of them had which psychotic symptoms?

We have added a table of the specific psychotic symptoms evinced by each participant to address this point (supplementary table 1)

– Line 503: EEG is a part of PSG

Noted and updated to reflect this (page 33)

– Which ICA components were removed? ECG related artefact or more?

The components removed were those scoring highest for ECG, EOG and EMG. We have added this to the methods (page 34)

– Why were artefacts removed only from N2 and N3, while powers spectra and sleep architecture were investigated for all sleep stages? This procedure could have artificially decrease the relative amount of N2 and N3 sleep. Furthermore, between-group differences in sleep architecture could be driven by differences in the amount of artifacts that were excluded.

Artefact removal was applied prior to quantitative EEG analyses to provide the highest quality epochs for analysis, to avoid contamination of our measures. However, as it was possible for the scorer to score all epochs by stage using the full PSG and video recordings, we investigated group differences in sleep architecture using the full night hypnogram. We did not observe any group differences in the amount of N2 or N3 epochs removed by our artefact removal (supplementary figure 1). Therefore, we believe that our approach to artefact removal will not bias our experimental findings

– Is spindle density calculated as number of events per minute?

We report spindle density as spindles per minute. We have clarified this in the methods (page 36)

– Line 627: what do the authors mean with "15 EEG measures"?

We refer to the full set of EEG derived measures we compared between groups. We had added a table laying out all the measures we use, Supplementary Table 4, which we hope will improve clarity for the reader

– Page 642: normally distributed across what? Participants (all or within a group), or across scalp sites?

For all statistical models, the model fit to the data uses particular distributional characteristics, e.g. linear models assume that the data have a normal distribution with a mean and variance which is estimated by the model, poisson models use the poisson distribution etc. The methods section lays out the parameters used for the Bayesian models, and stating the distribution used in the model is one of those parameters. We have changed the order of this section to clarify (page 39), and we also provide the R scripts used to fit these models.

Reviewer #3 (Recommendations for the authors):(1.) Line 103: remove "consolidation" from sleep dependent memory consolidation task.

Done.

(2.) Line 143: Introduce term PSD, before using abbreviation.

Done.

(3.) Line 153: The authors state that the spectral fingerprints of NREM EEG in young 22q11.2DS carriers differ from those of adult schizophrenia patients. Please briefly outline the differences (non-expert readers might be lost).

We have removed this sentence to clarify the Results section.